# TOWARDS THE THREE-PHASE DYNAMICS OF GENERALIZATION POWER OF A DNN

## ABSTRACT

This paper addresses the core challenge in the field of symbolic generalization[1], i.e., how to define, quantify, and track the dynamics of generalizable and non-generalizable interactions encoded by a DNN throughout the training process. Specifically, this work builds upon the recent theoretical achievement in explainable AI (Ren et al., 2024), which proves that the detailed inference patterns of DNNs can be strictly rewritten as a small number of AND-OR interaction patterns. Based on this, we propose an efficient method to quantify the generalization power of each interaction, and we discover a distinct three-phase dynamics of the generalization power of interactions during training. In particular, the early phase of training typically removes noisy and non-generalizable interactions and learns simple and generalizable interactions. The second and the third phases tend to capture increasingly complex interactions that are harder to generalize. Experimental results verify that the learning of non-generalizable interactions is the direct cause for the gap between the training and testing losses.

## 1 INTRODUCTION

Despite the rapid advancement of deep learning, a sophisticated theoretical understanding of the generalization power of deep neural networks (DNNs) remains elusive. In practice, the widely employed techniques for improving generalization are predominantly empirical, such as chain of thought (CoT) (Wei et al., 2022), data cleaning (Brown et al., 2020; Silcock et al., 2022), and large language model (LLM) alignment via reinforcement learning (Ouyang et al., 2022; Rafailov et al., 2023; Shao et al., 2024). Consequently, a gap persists between these practical methodologies and theoretical analyses of generalization (Foret et al., 2020; Li et al., 2018; Xiao et al., 2020).

Therefore, there is a clear tend towards more sophisticated and principled analysis of a DNN's generalization power, which is emerged in recent years. To this end, **can the generalization power of a DNN be attributed to the representation quality of the inference patterns in the DNN?**

**Background: symbolic generalization.** The emerging field of symbolic generalization[1] has served as a pioneering response to the above question and has garnered considerable attention (Liu et al., 2023; Ren et al., 2023b;c; Zhou et al., 2024) (see Section 4 for a survey). Within this field, (Ren et al., 2024) have proven a counterintuitive phenomenon: **the complex inference logic of a DNN can be accurately and comprehensively explained by a small set of symbolic interactions.** As illustrated both in Figure 1 and Figure 7, an interaction represents an AND logic or an OR logic among input variables that exerts a specific numerical influence on the model's output. For instance, a large language model (LLM) in Figure 1 encodes an AND interaction between $S = \{earth, revolves, sun\}$, which is only triggered when all the three words in $S$ are presented in the model input and exerts a numerical effect $I_T^{and} = 0.39$ that increases the LLM's output confidence in generating the token "year." Crucially, **(Ren et al., 2024) have proven such AND-OR interaction effects can accurately and comprehensively predict the DNN's outputs on exponentially massive samples, which ensures the faithfulness of regarding these interactions as the primitive inference patterns encoded by the DNN**.

**Our research**. In this paper, we address a fundamental and long-standing challenge in the field of symbolic generalization[1]: **how to rigorously quantify the generalization power of different**

---

[1]The entire theory system of symbolic generalization contains more than 20 papers. Representative studies in this direction have been surveyed in Section 4.

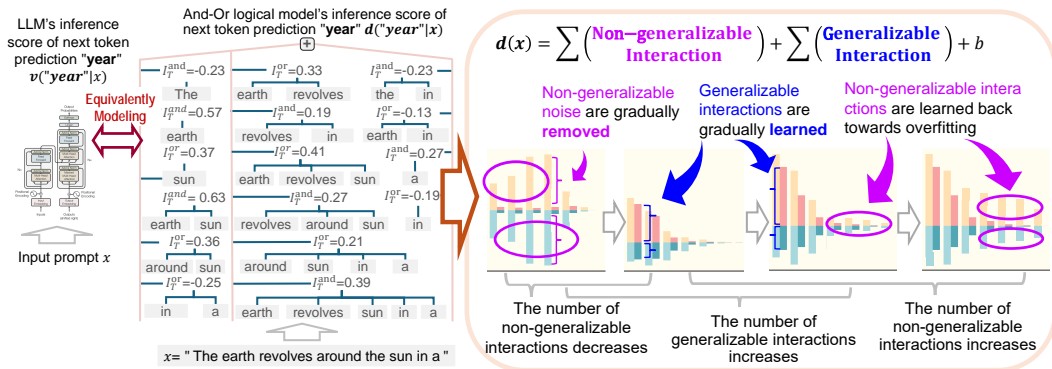

Figure 1: (Left) It has been proven that a small set of AND-OR interaction patterns are sufficient to mathematically represent all intricate inference patterns used by a DNN for inference. (Right) For each DNN, the change of its generalization power can be explained by the distinct three-phase dynamics of the generalization power of interactions encoded by it. **Please see Figure 7 in Appendix F for more AND-OR logical models that explain LLMs' outputs on different prompts.**

**interactions in a DNN and trace their evolution throughout the training process.** As Figure 7 shows, since the output of a DNN can be reformulated as the sum of all interaction effects, if most interactions learned by the DNN can also frequently appear in (be transferred to) unseen testing samples, then the DNN would exhibit a high testing accuracy, and vice versa. In other words, **the generalization power of interactions determines the generalization of the entire DNN.**

To this end, previous studies in the field of symbolic generalization remain limited to qualitative analysis of the complexity (Zhou et al., 2024) and robustness (Ren et al., 2023c) of interactions. People still lack a direct method for measuring the exact generalization power of each individual interaction, which has been a long-standing challenge for symbolic generalization for years.

Therefore, in this paper, we propose an efficient method to quantify the generalization power of each individual interaction. Instead of the computationally prohibitive approach of checking an interaction across all test samples, our method trains a baseline DNN on testing samples. All interactions, which can be transferred to the baseline DNN, are considered generalizable to testing samples. It is because each transferable interaction is also learned by the baseline DNN from testing samples as a primitive inference pattern.

The quantification of generalizable interactions allows us to **uncover the distinctive three-phase dynamics of generalization power of interactions through entire training process** (see Figure 3):

• Phase 1: Given a fully initialized DNN, in the early epochs of training, a large number of non-generalizable interactions are removed from the DNN, while a few generalizable interactions are gradually learned. These generalizable interactions are often simple and involve a small number of input variables, leading to the enhancement of the DNN's generalization power.

• Phase 2: The DNN continues to learn more interactions, but the newly learned interactions exhibit increasing complexity (*i.e.*, learning interactions between more input variables). These more complex interactions often have poorer generalization power. Thus, although the testing loss of the DNN continues to decreases, it tends to saturate due to the decreasing generalization power of the newly learned interactions.

• Phase 3: The DNN continues to learn additional interactions, but these newly learned interactions are often complex and difficult to apply to unseen data. Very few of these late-stage learned interactions represent simple, broadly generalizable patterns. Therefore, we can consider this phase revealing the overfitting of the DNN. During this phase, the gap between training and testing loss continues to widen. And the testing loss may even increase.

To further investigate how these non-generalizable interactions influence the DNN's performance, we propose to remove the effects of non-generalizable interactions from the DNN's output. We find that **the removal of non-generalizable interactions significantly reduces the gap between the training**

**loss and the testing loss, thereby effectively alleviating overfitting.** Notably, follow-up studies in Appendix K have also shown that penalizing non-generalizable interactions during the training process will improve the DNN's performance.

## 2 METHODOLOGY

### 2.1 PRELIMINARIES: AND-OR INTERACTIONS

Consider an input sample $\mathbf{x} = [x_1, x_2, \ldots, x_n]^T$ comprising $n$ input variables, where $N = \{1, 2, \ldots, n\}$ denotes the index set. Let $v(\mathbf{x}) \in \mathbb{R}$ denote the scalar output of the DNN. While $v(\mathbf{x})$ can be defined in various ways[2], we adopt the standard formulation widely used by (Ren et al., 2024; Zhou et al., 2024; Liu et al., 2023) to let $v(\mathbf{x})$ represent the classification confidence score for the ground-truth label $\mathbf{y}^*$ as

$$v(\mathbf{x}) = \log\left(\frac{p(\mathbf{y}^* \mid \mathbf{x})}{1 - p(\mathbf{y}^* \mid \mathbf{x})}\right). \tag{1}$$

**Problem setting:** A central challenge in the field of symbolic generalization[1] is to elucidate the complex inference patterns of a DNN (Li & Zhang, 2023b; Ren et al., 2024; Zhou et al., 2024). The basic idea is to utilize a logical model $d$ to explain all primitive inference patterns in the DNN $v$. The faithfulness of this explanation is contingent upon satisfying two fundamental requirements. **(1) Fidelity requirement:** The logical model $d$ must faithfully replicate the DNN's outputs over a comprehensively massive input set $\mathcal{Q}$. **(2) Conciseness requirement:** The logical model $d$ must contain parsimonious patterns to allow for succinct explanations, as specified below:

$$\forall \mathbf{x}' \in \mathcal{Q}, d(\mathbf{x}') = v(\mathbf{x}'), \text{ subject to } Complexity(d) \leq K, \tag{2}$$

where $K$ is the maximum complexity allowed for logical model $d$ to be considered concise.

Concretely, the logical model $d$ constructed below is designed to encode a set of AND-OR interaction patterns. *Please see a **video demonstration** for symbolic generalization in supplementary material. Furthermore, please see Figure 7 in Appendix F for more logical models that explain LLMs.*

$$d(\mathbf{x}') = \sum_{T \in \Omega_{\text{and}}} \underbrace{I_T^{\text{and}} \cdot \delta_{\text{and}}\left(\begin{smallmatrix} \mathbf{x}' \text{ triggers AND relation} \\ \text{between input variables in } T \end{smallmatrix}\right)}_{\text{an AND interaction between input variables in T}} + \sum_{T \in \Omega_{\text{or}}} \underbrace{I_T^{\text{or}} \cdot \delta_{\text{or}}\left(\begin{smallmatrix} \mathbf{x}' \text{ triggers OR relation} \\ \text{between input variables in } T \end{smallmatrix}\right)}_{\text{an OR interaction between input variables in T}} + b, \tag{3}$$

where $\mathbf{x}'$ denotes the input $\mathbf{x}$ where some input variables are masked[3] and, $b$ is a scalar bias.

*Each AND interaction* is represented by binary function $\delta_{\text{and}}\left(\begin{smallmatrix} \mathbf{x}' \text{ triggers AND relation} \\ \text{between input variables in } T \end{smallmatrix}\right) \in \{0, 1\}$. It returns 1 if all variables in $T$ are not masked in $\mathbf{x}'$; otherwise it returns 0. *Each OR interaction* is represented by a binary function $\delta_{\text{or}}\left(\begin{smallmatrix} \mathbf{x}' \text{ triggers OR relation} \\ \text{between input variables in } T \end{smallmatrix}\right) \in \{0, 1\}$. It returns 1 if any variables in $T$ are not masked in $\mathbf{x}'$; otherwise it returns 0. $I_T^{\text{and}}$ and $I_T^{\text{or}}$ are scalar weights.

**First, the fidelity requirement is fulfilled by the universal matching property established in Theorem 1.** This theorem demonstrates that a logical model with specific settings of $I_T^{\text{and}}$ and $I_T^{\text{or}}$ is able to accurately predict all outputs of the DNN across an exponential number of $2^n$ masked input states. The comprehensively massive sample set is then constructed as $\mathcal{Q} = \{\mathbf{x}_S \mid S \subseteq N\}$, which contains all $2^n$ masked instances of the input sample $\mathbf{x}$. $\mathbf{x}_S$ represents a masked input $\mathbf{x}$, in which input variables in $N \setminus S$ are masked[3].

**Theorem 1** (**universal matching property**, proven by (Chen et al., 2024)). *Given DNN $v$ and an input sample $\mathbf{x}$, let us set the scalar weights as $\forall T \subseteq N$, $I_T^{and} = \sum_{L \subseteq T}(-1)^{|T|-|L|}o_L^{and}$, $I_T^{or} = -\sum_{L \subseteq T}(-1)^{|T|-|L|}o_{N \setminus L}^{or}$ where $o_L^{and} = 0.5 \cdot v(\mathbf{x}_L) + \gamma_L$ and $o_L^{or} = 0.5 \cdot v(\mathbf{x}_L) - \gamma_L$, and set $b = v(\emptyset)$[4]. $\{\gamma_L\}$ are set of learnable parameters. The logical model can then accurately predict the*

---

[2]E.g., $v(\mathbf{x})$ can be set to the feature dimension of the ground-truth category prior to the softmax operation.

[3]The masking of input variable means substituting this variable with a baseline value. We set the baseline value of an input variable as the average value of this variable across different input samples. For NLP models, we use a specific embedding in (Cheng et al., 2024; Ren et al., 2024) to mask input tokens.

[4]$v(\emptyset)$ represents the network output when all input variables in $\mathbf{x}$ are masked.

*network outputs as follows, regardless of how* $\mathbf{x}$ *is randomly masked:*

$$\forall \mathbf{x}' \in \mathcal{Q}, \quad d(\mathbf{x}') = v(\mathbf{x}'). \tag{4}$$

**Computation of interactions.** Building upon the framework established in (Chen et al., 2024; Ren et al., 2023b), the parameters $\{\gamma_L\}$ are then optimized by minimizing a LASSO-regularized loss $\min_{\{\gamma_L\}} \|\mathbf{I}_{\text{and}}\|_1 + \|\mathbf{I}_{\text{or}}\|_1$ to obtain the sparsest interaction patterns, where we vectorize interactions as $\mathbf{I}_{\text{and}} = \left[I_{T_1}^{\text{and}}, \ldots, I_{T_{2^n}}^{\text{and}}\right]^T$, $\mathbf{I}_{\text{or}} = \left[I_{T_1}^{\text{or}}, \ldots, I_{T_{2^n}}^{\text{or}}\right]^T \in \mathbb{R}^{2^n}$. Please refer to Appendix A for the pseudocode for the computation of interactions and technical details.

**Second, the conciseness requirement is fulfilled by the sparsity[5] property of interactions.** Particularly, (Ren et al., 2024) have further proven that under three common conditions[6], DNNs can encode only a small number ($\mathcal{O}\left(n^p/\tau\right) \ll 2^{n+1}$) of AND-OR interactions with salient effects, while all other interactions have almost zero effects. Specifically, $\tau$ is a tiny threshold for saliency of interactions, and empirically, $p \in [0.9, 1.2]$.

The above two requirements enable us to construct a **concise logical model with a few salient AND-OR interactions in** $\Omega_{\text{and}}^{\text{Salient}} = \{S \mid |I_S^{\text{and}}| > \tau\}$ and $\Omega_{\text{or}}^{\text{Salient}} = \{S \mid |I_S^{\text{or}}| > \tau\}$, *which can faithfully match network outputs on all samples in* $\mathcal{Q}$. *The theoretical analysis in Appendix D and lots of empirical results from diverse DNNs (including LLMs) in both Appendix F and (Li & Zhang, 2023b; Ren et al., 2024; Zhou et al., 2024) support the the scientific rigor of the symbolic generalization.*

**The order of an interaction** $S$ represents the complexity of this interaction. The order is defined as the number of input variables involved in the interaction $\text{Order}_S = |S|$.

## 2.2 QUANTIFYING GENERALIZATION POWER OF INTERACTIONS

### 2.2.1 DEFINITION AND QUANTIFICATION

The above fidelity requirement and conciseness requirements have become foundational pillars for the emerging direction of symbolic generalization (Li & Zhang, 2023b; Ren et al., 2023c; Zhou et al., 2024). Crucially, studies in this direction introduce a paradigm shift in understanding the generalization behavior of DNNs. This is grounded in the observation that a DNN's predictive output can be mathematically decomposed into a linear combination of AND-OR interaction effects (see Figure 1). Consequently, a DNN's generalization power can be considered to be determined by the generalization power of the interactions in the DNN.

**Previous definition of the generalization power of interactions.** Let us first revisit the definition of generalization power of an interaction in (Zhou et al., 2024). Given a salient AND interaction $S$ extracted from an input $\mathbf{x}$ subject to $|I_S^{\text{and}}| > \tau$, if this interaction $S$ frequently occurs in the testing set and consistently making an effect on the classification of a category, then it is deemed to be generalizable; otherwise, it fails to generalize to testing samples. This criterion extends analogously to OR interactions. For instance, in a bird species classification task, suppose an AND interaction $S = \{red\ features, long\ beak\}$ consistently appears in both training and testing datasets and consistently increases the confidence in predicting the *"Flamingo"* class. In this case, the interaction $S$ demonstrates strong generalization power to unseen testing samples.

**Efficient quantification of the generalization power of interactions.** However, the above definition does not provide an efficient method for quantifying the generalization power of interactions. For example, in natural language processing tasks, this would necessitate an exhaustive search for specific interactions across numerous testing samples, leading to prohibitive computational costs.

Therefore, we propose an approximate and efficient method **that first quantifies the exact generalizability of each individual interaction.** Instead of performing an exhaustive search, we evaluate the transferability of individual interactions encoded by the DNN to a baseline DNN, denoted by $v^{\text{base}}$ and trained on the testing samples. Since all representations in the baseline DNN $v^{\text{base}}$ are acquired from the testing samples, all interactions, which are transferable to $v^{\text{base}}$, can be regarded as primitive

---

[5]In the field of symbolic generalization, the sparsity is defined as the state that almost all interactions have negligible values with only very few interactions having salient values. Please see Appendix B for details.

[6](Ren et al., 2024) have demonstrated that the sparsity of interactions can be ensured through three common conditions for smooth inferences in DNNs on randomly masked samples. See Appendix B for more details.

inference patterns inherent in these testing samples, thus strongly suggesting their generalization power. Please refer to Appendix C for ablation study.

To elucidate this approach, let us consider an extracted[7] salient AND interaction $S$, *s.t.* $|I_S^{\text{and}}| > \tau$. To this end, the interaction $S$ is deemed to represent a generalizable pattern within the testing set if the baseline DNN $v^{\text{base}}$ concurrently identifies the AND interaction $S$ to have salient effect (*i.e.*, $|I_{S,v^{\text{base}}}^{\text{and}}| > \tau$) and the AND interaction $S$ exerts a consistent directional influence on the classification of $\mathbf{x}$ (*i.e.*, $I_{S,v^{\text{base}}}^{\text{and}} \cdot I_S^{\text{and}} > 0$). The same principle applies to OR interactions. Accordingly, the generalization power of an AND/OR interaction is assessed by the following binary metrics:

$$
\begin{aligned}
\mathcal{G}_{S,v^{\text{base}}}^{\text{and}} &= \mathbf{1}(|I_{S,v^{\text{base}}}^{\text{and}}| > \tau) \cdot \mathbf{1}(I_S^{\text{and}} \cdot I_{S,v^{\text{base}}}^{\text{and}} > 0) \in \{0, 1\}, \\
\mathcal{G}_{S,v^{\text{base}}}^{\text{or}} &= \mathbf{1}(|I_{S,v^{\text{base}}}^{\text{or}}| > \tau) \cdot \mathbf{1}(I_S^{\text{or}} \cdot I_{S,v^{\text{base}}}^{\text{or}} > 0) \in \{0, 1\},
\end{aligned}
\tag{5}
$$

where $\mathbf{1}(\cdot)$ is a binary indicator function that outputs 1 only when the condition is satisfied.

### 2.2.2 VERIFYING PREVIOUS FINDINGS ON THE GENERALIZATION POWER

The above metrics $\mathcal{G}_{S,v^{\text{base}}}^{\text{and}}$ and $\mathcal{G}_{S,v^{\text{base}}}^{\text{or}}$ allow us to further verify and elucidate decisive factors behind the classical empirical finding in symbolic generalization[1] (Zhou et al., 2024), *i.e.*, low-order interactions generally exhibit stronger generalization power compared to high-order interactions, which was observed by (Zhou et al., 2024) in experiments[8] and lacked rigorous quantification and analysis.

Therefore, although the above claim regarding the low generalization power of high-order interactions has been widely cited by (Chen et al., 2024; Cheng et al., 2024), this assertion still requires more rigorous verification by identifying the generalization power of each specific salient interaction.

Specifically, to assess the generalization power of interactions[7] at distinct orders[9], we utilize four distinct metrics: (1) $\mathbb{I}_{\text{total}}^{(m),+}$ denotes the aggregate effect strength of all salient interactions of the $m$-th order with positive effect; (2) $\mathbb{I}_{\text{total}}^{(m),-}$ denotes the aggregate effect strength of all salient interactions of the $m$-th order with negative effect; (3) $\mathbb{I}_{\text{gen}}^{(m),+}$ denotes the aggregate effect strength of all generalizable salient interactions with positive effect; (4) $\mathbb{I}_{\text{gen}}^{(m),-}$ denotes the aggregate effect strength of all generalizable salient interactions with negative effect. The metrics are calculated as

$$
\mathbb{I}_{\text{total}}^{(m),-} = \sum_{\substack{\text{type} \in \\ \{\text{and, or}\}}} \sum_{\substack{S \in \Omega_{\text{type}}^{\text{Salient}}: \\ |S|=m}} \max(I_S^{\text{type}}, 0), \quad
\mathbb{I}_{\text{total}}^{(m),+} = \sum_{\substack{\text{type} \in \\ \{\text{and, or}\}}} \sum_{\substack{S \in \Omega_{\text{type}}^{\text{Salient}}: \\ |S|=m}} \min(I_S^{\text{type}}, 0),
\tag{6}
$$

$$
\mathbb{I}_{\text{gen}}^{(m),-} = \sum_{\substack{\text{type} \in \\ \{\text{and, or}\}}} \sum_{\substack{S \in \Omega_{\text{type}}^{\text{Salient}}: \\ |S|=m}} \max(I_S^{\text{type}} \cdot \mathcal{G}_S^{\text{type}}, 0), \quad
\mathbb{I}_{\text{gen}}^{(m),+} = \sum_{\substack{\text{type} \in \\ \{\text{and, or}\}}} \sum_{\substack{S \in \Omega_{\text{type}}^{\text{Salient}}: \\ |S|=m}} \min(I_S^{\text{type}} \cdot \mathcal{G}_S^{\text{type}}, 0).
\tag{7}
$$

To this end, we conducted experiments to analyze the strength of generalizable and non-generalizable interactions in comparison across different interaction orders. We trained VGG-19 and ResNet-101 on the Tiny-ImageNet dataset, and VGG-16 on the CUB-200-2011 dataset. We also trained Bert-large and Bert-medium on the SST-2 dataset and Bert-large on the AG News dataset. Please refer to Appendix E for further experimental details. Figure 2 demonstrates that high-order interactions consistently exhibited lower generalization power than low-order interactions in different cases.

## 3 EVALUATION

### 3.1 THREE-PHASE DYNAMICS OF GENERALIZATION POWER OF INTERACTIONS

Based on the new generalization power metrics $\mathbb{I}_{\text{total}}^{(m),+}$, $\mathbb{I}_{\text{total}}^{(m),-}$, $\mathbb{I}_{\text{gen}}^{(m),+}$, $\mathbb{I}_{\text{gen}}^{(m),-}$ proposed in Section 2.2, this subsection investigates the evolution of generalization power of interactions throughout the

---

[7]Interaction extraction is introduced in both Section 2.1 and the pseudocode in Appendix A.

[8]Evidence from two experiments substantiates this claim: (1) It is observed that, when more label noise is introduced, overfitted DNNs tend to model more high-order interactions compared to standard DNNs trained on clean data. (2) It is demonstrated that high-order interactions typically show worse robustness to input noise.

[9]As explained in Section 2.1, the order of an interaction refers to the number of input variables it involves.

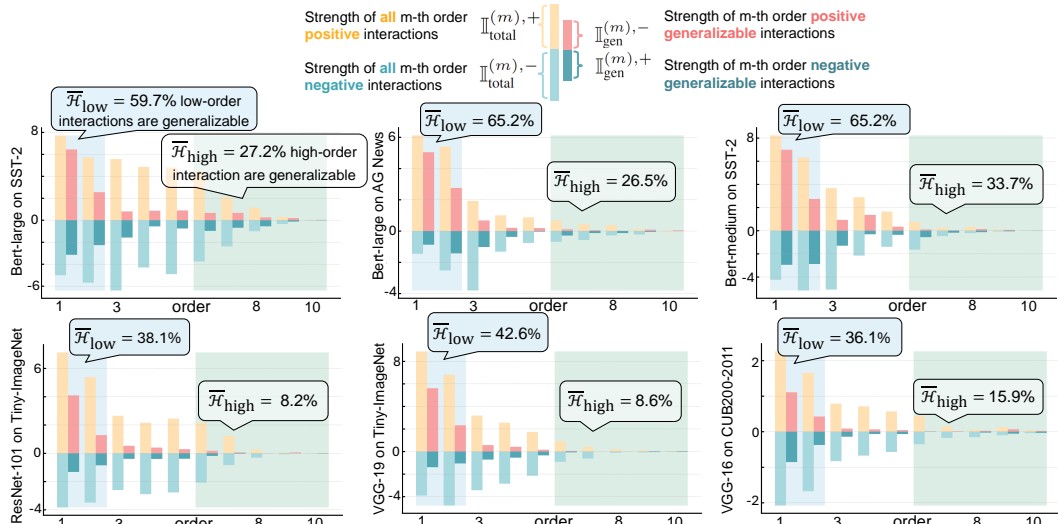

Figure 2: Comparison of generalization power between low-order interactions and high-order interactions. High-order interactions generally exhibit weaker generalization power than low-order interactions. Considering up to 10th order, we report the ratio of generalizable interactions of low orders as $\overline{\mathcal{H}}_{\text{low}} = \sum_{m=1}^{2}(|\mathbb{I}_{\text{gen}}^{(m),+}| + |\mathbb{I}_{\text{gen}}^{(m),-}|) / \sum_{m=1}^{2}(|\mathbb{I}_{\text{total}}^{(m),+}| + |\mathbb{I}_{\text{total}}^{(m),-}|)$ and the ratio of generalizable interactions of high orders as $\overline{\mathcal{H}}_{\text{high}} = \sum_{m=6}^{10}(|\mathbb{I}_{\text{gen}}^{(m),+}| + |\mathbb{I}_{\text{gen}}^{(m),-}|) / \sum_{m=6}^{10}(|\mathbb{I}_{\text{total}}^{(m),+}| + |\mathbb{I}_{\text{total}}^{(m),-}|)$.

entire training process, which is recognized as a long-standing challenge in the field of symbolic generalization[1]. Specifically, we propose the following metric $\overline{\mathcal{H}}$ to measure the average generalization power of all salient interactions on an input sample $\mathbf{x}$ as follows:

$$\overline{\mathcal{H}} = \sum_{m} \left( |\mathbb{I}_{\text{gen}}^{(m),+}| + |\mathbb{I}_{\text{gen}}^{(m),-}| \right) / \sum_{m} \left( |\mathbb{I}_{\text{total}}^{(m),+}| + |\mathbb{I}_{\text{total}}^{(m),-}| \right). \tag{8}$$

We measure the number of interactions $\overline{\mathcal{N}}$ as the number of top-ranked salient interactions that account for 80.0% of the total interaction strength. Specifically, the number of AND interactions, denoted as $\overline{\mathcal{N}}_{\text{and}}$, can be determined as follows:

$$\overline{\mathcal{N}}_{\text{and}} = \underset{\Omega \subseteq 2^N}{\operatorname{argmin}} |\Omega| \quad \text{s.t.} \quad \sum_{S \in \Omega} |I_S^{\text{and}}| \geq 0.8 \cdot \sum_{S \subseteq N} |I_S^{\text{and}}|. \tag{9}$$

where $2^N = \{S \subseteq N\}$. The number of OR interactions, denoted as $\overline{\mathcal{N}}_{\text{or}}$, can be quantified through the same approach. Therefore, the overall interaction number $\overline{\mathcal{N}}$ is calculated as

$$\overline{\mathcal{N}} = \overline{\mathcal{N}}_{\text{and}} + \overline{\mathcal{N}}_{\text{or}}. \tag{10}$$

**Three-phase dynamics.** Figure 3 shows that for most DNNs, the distribution of interactions follows a three-phase dynamics in terms of the generalization power $\overline{\mathcal{H}}$ and the interaction number $\overline{\mathcal{N}}$, which can be summarized as:

*Finding 1: During the training of all DNNs, the number of interactions always exhibits **a regularity of first decreasing and then increasing**.*
*Finding 2: The average generalization power of interactions always demonstrates **a regularity of first increasing and then decreasing**.*
*Finding 3: The three-phase dynamics of interactions is temporally **aligned with and well explains decisive factors behind the curve of the training-testing loss gap**. Detailed dynamics is as follows.*

• **Before Training**: Given a DNN with fully initialized parameters, although all interactions extracted from the initialized DNN represent random noise patterns, statistically, the interactions exhibit a specific spindle-shaped distribution. In words, the model primarily encodes middle-order interactions, while very high-order and very low-order interactions are rarely modeled. Furthermore, the effects of the positive interactions and the negative interactions often tend to cancel out. This phenomenon has been demonstrated in previous study (Cheng et al., 2025).

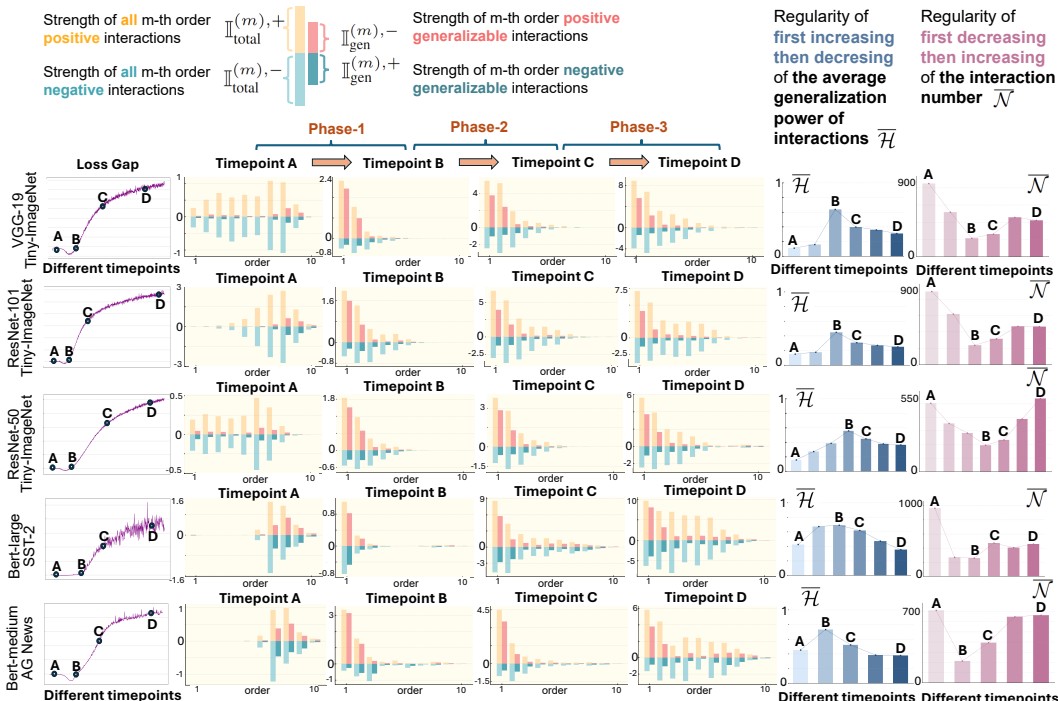

Figure 3: We visualize the change of interactions through the entire training process and track the change of the average generalization power of interactions $\overline{\mathcal{H}}$ and the number of interactions $\overline{\mathcal{N}}$. The dynamics of quantity and the average generalization power of interactions explains the change of the loss gap. Please see Appendix E for the training loss and the testing loss.

At this stage, we observe that most interactions cannot transfer to the baseline DNN. This corroborates that interactions encoded in an initialized DNN are noise patterns.

• **Phase 1**: In the early epochs of training, the average generalization power of interactions $\overline{\mathcal{H}}$ usually increases, and the number of generalizable interactions $\overline{\mathcal{N}}$ usually decreases. The training primarily eliminates noise in the spindle-shaped distribution while simultaneously learning a few low-order interactions. During this phase, the DNN mainly learns these low-order interactions, which tend to generalize better to the baseline DNN as they capture simple, meaningful patterns. Meanwhile, high-order interactions initially modeled by the DNN are gradually eliminated.

According to Equation (15), we can conclude that the classification confidence depends jointly on the number of salient interactions and the average generalization power of these interactions. Therefore, the emergence of generalizable low-order interactions and the elimination of non-generalizable high-order interactions characterize the healthiest stage of training.

• **Phase 2**: As training progresses, the average generalization power of interactions $\overline{\mathcal{H}}$ gradually decreases, and the number of interactions $\overline{\mathcal{N}}$ usually increases. Both generalizable (mainly low-order) interactions and non-generalizable (mainly mid-to-high-order) interactions are learned by the DNN simultaneously. Since the initial noise interactions have been removed after the first phase, in the second phase the DNN progressively learns interactions of increasing order. Notably, we find that high-order interactions are generally less likely to be generalized to testing samples than lower-order interactions according to Section 2.2.2. Thus, two phenomena co-occur in the second phase: (1) the DNN continues to learn additional interactions; (2) the generalization power of newly learned interactions steadily decreases.

Consequently, the second phase faces a dilemma. Although the continuous learning of new interactions provides meaningful features for classification, the diminishing generalization power of these interactions introduces a risk of overfitting. This observation is further corroborated by the training curves: The rate of decrease of the training loss outpaces that of the testing loss, resulting in a widening gap that indicates potential risk of overfitting of the model to the training data. This phe-

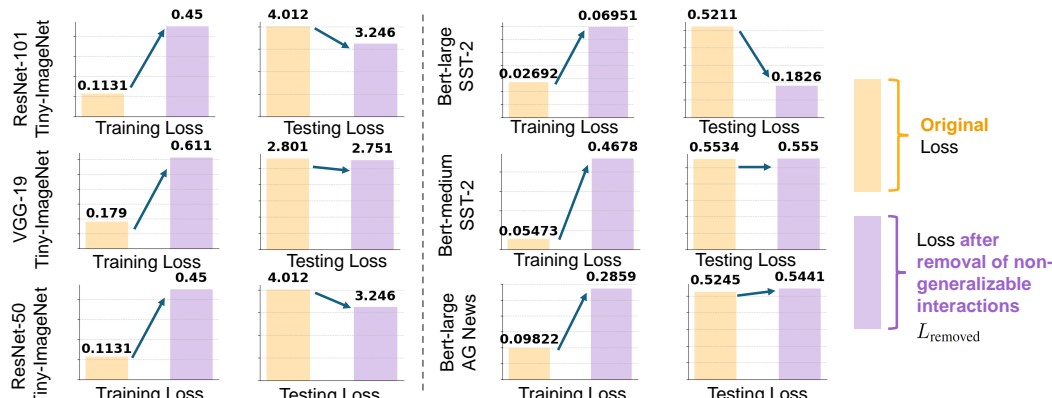

Figure 4: The change of the cross-entropy loss when we remove non-generalizable interactions from the network output $v_{\text{removed}}(\mathbf{x})$. The removal of non-generalizable interactions significantly increases the training loss, which restores the training loss that was inappropriately depressed by overfitted representations. In comparison, the testing loss either decreases significantly or does not change a lot.

nomenon may arise because non-generalizable interactions contribute exclusively to the classification of training samples and fail to aid in the classification of unseen testing samples.

• **Phase 3**: During the final training stage, DNNs primarily acquire non-generalizable (mainly mid-to-high-order) interactions, while it learns minimal generalizable (predominantly low-order) interactions. As a matter of fact, low-order interactions have already been thoroughly captured during the preceding two phases. Meanwhile, these emergent high-order interactions typically demonstrate poor generalization to unseen samples.

We hypothesize that these high-order interactions represent potential sources of overfitting. While generalizable interactions facilitate effective classification of unseen testing samples, non-generalizable interactions widen the gap between training and testing loss.

## 3.2 INFLUENCE OF NON-GENERALIZABLE INTERACTIONS ON THE CLASSIFICATION LOSS

In addition to the three-phase dynamics of interactions' generalization power, in this subsection, we propose an alternative perspective to examine if the non-generalizable[10] interactions are truly responsible for the overfitting of the DNN. The basic idea is that if we remove the effects of non-generalizable interactions from the scalar output of the DNN, then we can test **whether the removal of non-generalizable interactions will reduce the training-testing loss gap.** To accomplish this, let us rewrite the scalar output of the DNN in Equation (3) as (see Appendix H for proof)

$$
\begin{aligned}
v(\mathbf{x}) = &\sum_{\text{type}\in\{\text{and, or}\}} \sum_{S\in\Omega_{\text{type}}^{\text{Salient}}} I_S^{\text{type}} \cdot G_S^{\text{type}} \cdot \delta_{\text{type}}\left(\begin{smallmatrix}\mathbf{x}\text{ triggers type relation}\\ \text{between input variables in } T\end{smallmatrix}\right) \quad \textit{*generalizable interactions}\\
&+ \sum_{\substack{\text{type}\in\\\{\text{and, or}\}}} \sum_{S\in\Omega_{\text{type}}^{\text{Salient}}} I_S^{\text{type}} \cdot (1 - G_S^{\text{type}}) \cdot \delta_{\text{type}}\left(\begin{smallmatrix}\mathbf{x}\text{ triggers type relation}\\ \text{between input variables in } T\end{smallmatrix}\right) \quad \textit{*non-generalizable interactions}\\
&+ \sum_{\text{type}\in\{\text{and, or}\}} \sum_{S\in\Omega_{\text{type}}^{\text{Non-Salient}}} I_S^{\text{type}} \cdot \delta_{\text{type}}\left(\begin{smallmatrix}\mathbf{x}\text{ triggers type relation}\\ \text{between input variables in } T\end{smallmatrix}\right) + b \quad \textit{*negligible non-salient interactions},
\end{aligned}
$$
(11)

where $\Omega_{\text{type}}^{\text{Non-Salient}} = \{S : |I_S^{\text{or}}| \leq \tau\}$ represents the set of non-salient interactions.

---

[10] When interaction $S$ does not satisfy the generalization condition in Equation (5), *i.e.*, when interaction S does not have a salient effect or does not exert a consistent directional influence. And we have $\mathcal{G}_{S,v^{\text{base}}}^{\text{type}} = 0$.

Then, the probability of classifying the input $\mathbf{x}$ to the ground-truth label $\mathbf{y}^*$ after the removal of non-generalizable interactions can be derived from Equation (1) as follows:

$$p_{\text{removed}}(\mathbf{y}^*|\mathbf{x}) = \text{sigmoid}(v_{\text{removed}}(\mathbf{x}))$$

$$\text{subject to} \quad v_{\text{removed}}(\mathbf{x}) = v(\mathbf{x}) - \sum_{\text{type}\in\{\text{and, or}\}} \sum_{S\in\Omega_{\text{type}}^{\text{Salient}}} I_S^{\text{type}} \cdot (1 - G_S^{\text{type}})). \qquad (12)$$

Then, given the revised classification probability $p_{\text{removed}}(\mathbf{y}^*|\mathbf{x})$, we can compute the new cross-entropy loss as: $L_{\text{removed}} = -\sum_{\mathbf{x}} \log p_{\text{removed}}(\mathbf{y}^*|\mathbf{x})$.

We conducted experiments to train VGG-19, ResNet-50, and ResNet-101 on the Tiny-ImageNet dataset, and Bert-large and Bert-medium on the SST-2 dataset, and Bert-large the AG News dataset. We computed the standard cross-entropy loss using the original classification probability $p(\mathbf{y}^*|\mathbf{x})$, as well as a modified cross-entropy loss using the revised probability $p_{\text{removed}}(\mathbf{y}^*|\mathbf{x})$. Please see Appendix J for details. As shown in Figure 4, **the removal of non-generalizable interactions significantly reduced the training-testing loss gap.** This removal primarily increased the training loss while significantly decreasing or having minimal impact on the testing loss. These results suggested that the removed non-generalizable interactions were highly correlated with the overfitting of the DNN, **as they exclusively contributed to training samples without generalizing to unseen data.**

## 4   RELATED WORK: THEORY SYSTEM OF SYMBOLIC GENERALIZATION

*Symbolic explanation: a seemingly impossible task.* Studies (Deng et al., 2022; Li & Zhang, 2023b; Ren et al., 2023b) show DNN inference can be reformulated via symbolic interactions between inputs. Though counterintuitive, (Ren et al., 2023a) proved a logical model based on such interactions can approximate DNN outputs on masked inputs. Further, (Ren et al., 2024) found DNNs encode only a limited set of salient interactions under common conditions[6]. Complementing this, (Chen et al., 2024) developed a method to extract generalizable interaction patterns consistent across models. These works form a mathematical basis for interpreting DNNs via variable interactions.

*Characterizing DNN representational capacity.* Interaction theory also serves as a tool to analyze DNN representation quality. (Ren et al., 2021) showed adversarial attacks target complex interactions, with robustness decaying exponentially as interaction complexity grows. (Ren et al., 2023c) proved Bayesian networks (Pearl, 1985) inherently struggle to model complex interactions. (Deng et al., 2021) identified a DNN limitation: they capture very simple or highly complex interactions but falter with intermediate ones. (Zhang et al., 2021) found dropout improves generalization by modulating interactions. (Zhou et al., 2024) verified simpler interactions generalize better. (Liu et al., 2023) gave theoretical evidence that DNNs find complex interactions harder to learn.

However, towards the core issue of the generalization power of interactions, previous studies in the field of symbolic generalization all failed to quantify the generalization power of a specific interaction. To this end, we propose to use a baseline DNN to identify generalizable interactions, which breaks through the computational efficiency bottleneck. In this way, we discover a distinctive three-phase dynamics in how generalization power evolves during training. Significantly, our research connects DNN's overfitting to the learning of non-generalizable interactions. Our discoveries establish a clear causal relationship between interaction patterns and a DNN's generalization power.

## 5   CONCLUSION

In this paper, we focus on the core issues in the field of symbolic generalization, *i.e.*, how to quantify generalization power of each individual interaction in a DNN and how to utilize the evolution of generalization power of interactions to diagnose the training process of a DNN. Specifically, we propose an efficient method to quantify the generalization power of each individual interaction learned by the DNN, so that we can use generalization power of all interactions to interpret the generalization power of the entire DNN. Our analysis reveals a three-phase dynamics in the generalization power of interactions, which well aligns the change of the DNN's training-testing loss gap. In the beginning, most non-generalizable interactions are eliminated, and the network primarily models simple, generalizable interactions. In the intermediate phase, the network begins to capture increasingly less

generalizable interactions, and by the final phase, it predominantly learns entirely non-generalizable ones. Our experiments demonstrate that the learning of non-generalizable interactions is a direct cause for the gap between the training loss and the testing loss.

**Practical value.** Our experiments have demonstrated that removing non-generalizable interactions can reduce the training-testing loss gap. Besides, follow-up studies in Appendix K have also shown that penalizing the strength of non-generalizable interactions $|I_S^{\text{NoneG}}|$ along with the minimization of the classification loss during the training process substantially improves the DNN's performance.

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

APPENDIX

This appendix provides detailed information supporting our main paper. Appendix A describes the details of computing interactions. Appendix B discusses three common conditions for the sparsity of interactions. Appendix C provides ablation study on the number of baseline DNNs. Appendix E contains experimental details. Appendix H presents proof of Equation 12 in the main paper. Appendix J details our experiments on removing non-generalizable interactions from model outputs, including sample selection methods in multiple datasets. Appendix L describes the computing resources and experimental time consumed for computing in this study.

# A  COMPUTATION OF INTERACTIONS

## A.1  SETTING OF INTERACTION COMPUTATION

For an input sample $\mathbf{x}$ with $n$ distinct input variables, the total number of subsets to evaluate scales as $2^n$, rendering exhaustive computation intractable when each pixel (in vision) or each token (in language) is treated as an individual variable. To mitigate this, we adopt the protocol of prior work (Li & Zhang, 2023b; Liu et al., 2023) by restricting the interaction analysis to exactly 10 variables, as detailed below.

**Image task.** We conduct experiments on image classification using convolutional networks (ResNet-50, ResNet-101, VGG-19). From a chosen intermediate feature map of each model (the first Relu layer), we partition the spatial dimensions into an $5 \times 5$ grid and sample 10 patches to form the set $N$. We set the baseline value of each input variable to its empirical mean across the dataset. For any subset $S \subseteq N$, a masked input $\mathbf{x}_S$ is generated by replacing all patches in $N \setminus S$ with $b$, thereby enabling efficient estimation of interaction strengths.

**Natural language task.** For text classification task, we employ BERT-Medium and BERT-Large architectures. For each input sentence, stop words, punctuation, and non-alphanumeric symbols removed. In this way, we then randomly select 10 tokens with semantic meanings to constitute $N$. To construct each masked sample, we follow the masking embedding scheme of (Cheng et al., 2024; Ren et al., 2024): for BERT models, every token in $N \setminus S$ is substituted with the pre-trained [MASK] embedding vector (ID=103), yielding $\mathbf{x}_S$ and facilitating computation of higher-order token interactions.

---

**Algorithm 1** Extract Interactions

> **Input:** Trained DNN $v(\cdot)$; input sample $\mathbf{x}$ (features indexed $1, \ldots, n$; let $N = \{1, \ldots, n\}$ be the set of these indices).
> **Constants assumed: number of top interactions $k$.**
> **Output:** Interaction weights $I_S^{\text{and}}, I_S^{\text{or}}$ for all $S \subseteq N$; bias $b$; sets of top-k salient interactions $\Omega_{\text{and}}^{\text{Salient}}, \Omega_{\text{or}}^{\text{Salient}}$.
> **for all** $S \subseteq N$ **do**
>     Generate masked input $\mathbf{x}_S$ and the corresponding model output $v(\mathbf{x}_S)$.
> **end for**
> **for all** $S \subseteq N$ **do**
>     Determine $I_S^{\text{and}}, I_S^{\text{or}}$ by solving $\min_{\{\delta_S, \gamma_S\}} \sum_{S \subseteq N} (|I_S^{\text{and}}| + |I_S^{\text{or}}|)$
> **end for**
> Determine the bias term $b$ as the network's output for the empty set of features: $b \leftarrow v(\emptyset)$.
> **for all** $S \subseteq N$ **do**
>     Filter $|I_S^{\text{and}}|$ and $|I_S^{\text{or}}|$ to identify the most salient $k$ interactions to construct $\Omega_{\text{and}}^{\text{Salient}}$ and $\Omega_{\text{or}}^{\text{Salient}}$
> **end for**
> **return** the computed interactions $I_S^{\text{and}}, I_S^{\text{or}}$, the bias $b$, and the sets of salient interactions $\Omega_{\text{and}}^{\text{Salient}}, \Omega_{\text{or}}^{\text{Salient}}$.

---

## A.2 Presucode for computation of interactions

Additionally, drawing upon insights from (Chen et al., 2024; Ren et al., 2024; Li & Zhang, 2023a), the presence of noise $\sigma_S$ in the model output $v(\mathbf{x})$ can significantly impede the accurate computation of interactions. Following the approach outlined in (Li & Zhang, 2023a) to address this challenge, we remove the noise $\sigma_S$ from the output $v(\mathbf{x})$ to enable the extraction of AND-OR interactions. More specifically, the decomposition $v(\mathbf{x}_S) = o_S^{\text{and}} + o_S^{\text{or}} + \sigma_S$ is reformulated such that $o_S^{\text{and}} = 0.5 \cdot (v(\mathbf{x}_S) - \sigma_S) + \gamma_S$ and $o_S^{\text{or}} = 0.5 \cdot (v(\mathbf{x}_S) - \sigma_S) - \gamma_S$. Consequently, the parameters $\{\gamma_S\}$ and $\{\sigma_S\}$ are optimized within a LASSO-like loss framework. This process is designed to enforce sparsity and mitigate the influence of noise on the model output, as detailed in pseudocode 1.

## B Three common conditions for the sparsity of interactions

In the field of symbolic generalization, the sparsity is defined as the state that almost all interactions have negligible values with only very few interactions having salient values. And (Ren et al., 2024) have proven the sparsity of interactions under three common conditions as follows.

**Condition 1: finite-Order interactions**

The network's output, denoted by a function $v$, can be fully characterized by interactions involving at most $M$ input variables. This implies that any interaction effects $I(S)$ for subsets of input variables $S \subseteq \{1, \ldots, n\}$ where the size of the subset $|S| > M$ are zero: $I(S) = 0$ for all $S$ such that $|S| > M$. Equivalently, in a Taylor expansion of the function $v$, all terms corresponding to mixed partial derivatives of order $M + 1$ or higher vanish. Specifically, for any point $b \in \mathbb{R}^n$ and any non-negative integers $\kappa_1, \ldots, \kappa_n$ such that $\sum_{i=1}^{n} \kappa_i \geq M + 1$, we have: $\left. \frac{\partial^{\kappa_1 + \cdots + \kappa_n} v}{\partial x_1^{\kappa_1} \ldots \partial x_n^{\kappa_n}} \right|_{x=b} = 0$. Such a constraint limits model complexity to lower-order effects, aligning with observations in some large models where very high-order interaction strengths are minimal, thereby directly supporting the premise of sparse high-order interactions.

**Condition 2: monotonicity of average network output**

Let $v(x_S)$ be the network's output when only the features in subset $S$ are revealed (and others are masked or set to a baseline $x_\varnothing$). Define the average increase in network output when exactly $m$ features are revealed as: $\bar{u}(m) = \mathbb{E}_{S \subseteq \{1, \ldots, n\}: |S| = m} \left[ v(x_S) - v(x_\varnothing) \right]$. This condition requires that the average network output is monotonically non-decreasing with the number of revealed features. That is, for any $m' \leq m$: $\bar{u}(m') \leq \bar{u}(m)$. Average monotonic behavior suggests feature interactions have underlying simplicity. Models dominated by many complex high-order interactions would unlikely show such stability. This predictable response pattern indicates sparse, lower-order interactions drive outputs rather than numerous erratic high-order effects, indirectly showing the interaction structure isn't arbitrarily complex.

**Condition 3: Polynomial lower bound on average output under masking**

There exists a positive constant $p > 0$ such that for any number of revealed features $m'$ and $m$ with $m' \leq m$, the average network output $\bar{u}(m')$ (as defined under Condition 2) is lower-bounded as follows: $\bar{u}(m') \geq \left( \frac{m'}{m} \right)^p \bar{u}(m)$. The polynomial bound on output degradation when masking features suggests an interaction structure without dense high-order terms. If complex high-order interactions were critical, feature removal would cause more drastic drops. This supports the sparsity hypothesis for impactful high-order interactions.

## C Ablation study towards the number of baseline DNNs

When we delve deeper into the problem of interaction generalization power, we observe that there is no clear boundary between generalizable and non-generalizable interactions. Consequently, the quantification of interaction generalization power is caught in the following dilemma.

(1) On one hand, we can only guarantee that all interactions modeled by a baseline DNN are learned from the testing samples. However, we cannot ensure that the baseline DNN models all possible

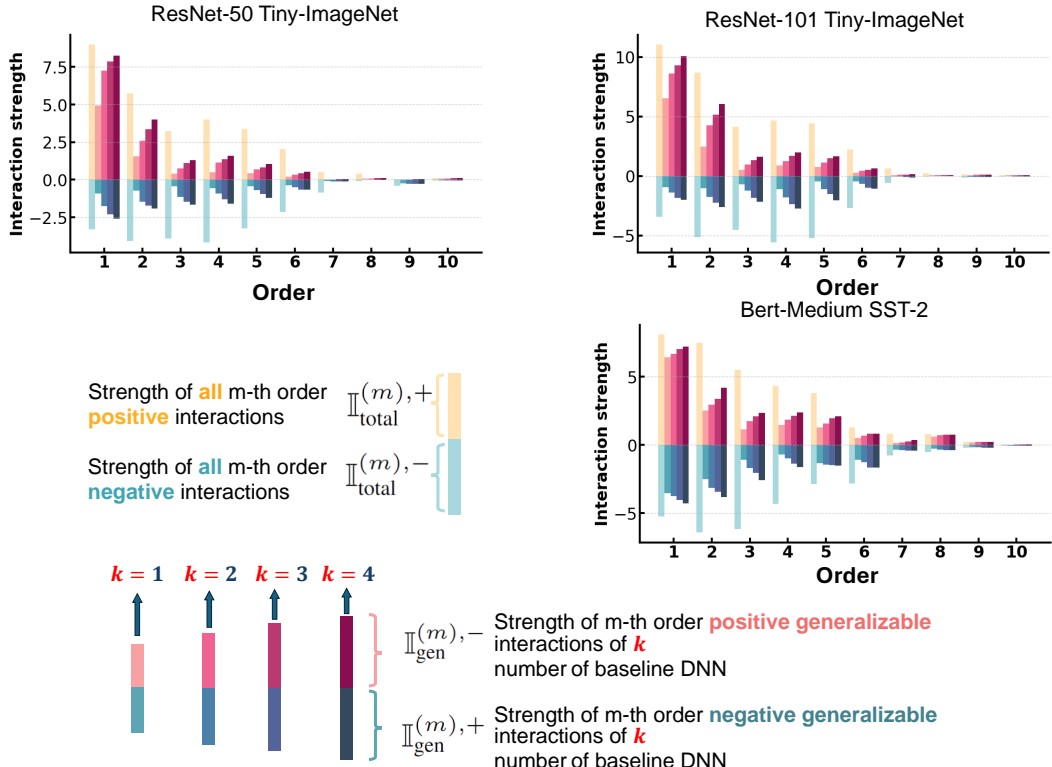

Figure 5: Ablation results of distribution of generalizable interactions with varying numbers of baseline DNNs. The distributions of interactions across different orders demonstrate a positive relationship wherein the strength of interactions increases proportionally with increasing values of $k$

interactions. This issue necessitates training multiple baseline DNNs to cover as many different generalizable interactions as possible.

(2) On the other hand, given a group of baseline DNNs $v_1^{\text{base}}, v_2^{\text{base}}, \ldots, v_K^{\text{base}}$, each generating a set of salient AND-OR interactions $\Omega_{\text{and/or},k}^{\text{Salient}}$, determining the generalization power of a target interaction becomes a fuzzy problem. If an interaction can only be generalized to one baseline DNN out of all $K$ baseline DNNs, whether this interaction is truly a generalizable one remains an open problem.

Therefore, we conducted ablation experiments to illustrate the generalizable interactions that were identified by using different numbers of baseline DNNs. Given $k$ baseline DNNs, a generalizable AND/OR interaction $S$ was defined by its ability to generalize to any one baseline DNN, as follows.

$$\hat{\mathcal{G}}_S^{\text{type}}(k) = \mathbf{1}\underbrace{\left(\mathcal{G}_{S,v_1^{\text{base}}}^{\text{type}} = 1 \vee \mathcal{G}_{S,v_2^{\text{base}}}^{\text{type}} = 1 \vee \cdots \vee \mathcal{G}_{S,v_k^{\text{base}}}^{\text{type}} = 1\right)}_{S \text{ generalized to any one of the } \textit{baseline DNNs}} \in \{0,1\} \qquad (13)$$

where type $\in \{\text{and, or}\}$ corresponds to AND/OR interactions, and $\mathbf{1}(\cdot)$ is a binary indicator function that outputs 1 if the condition is satisfied and 0 otherwise. Figure 5 shows the distribution of generalizable interactions and that of non-generalizable interactions under different numbers of baseline DNNs for different models. As evident from the figure, the proportion of generalizable interactions increases consistently with the number of baseline DNNs across all interaction orders. This observation aligns with our theoretical understanding that employing more baseline DNNs enhances the detection capability for generalizable interactions. These results empirically validate our approach of using multiple baseline DNNs to comprehensively capture the spectrum of generalizable interactions, thereby addressing the dilemma outlined earlier. Notably, it is evident that selecting $k = 2$ for our experiments offers a favorable balance between effectiveness and computational efficiency.

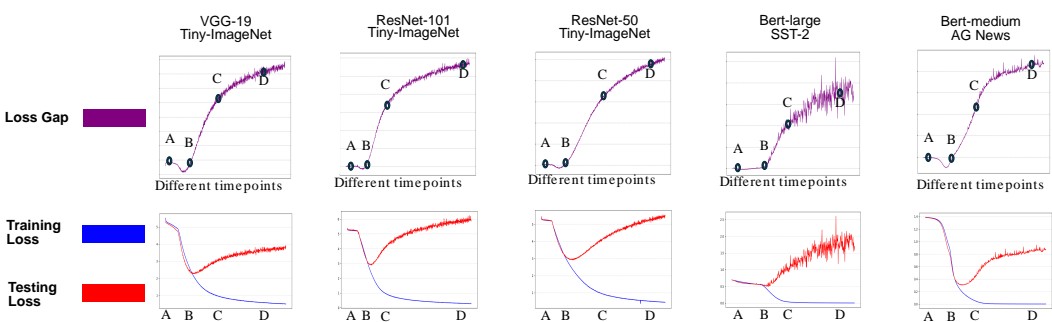

Figure 6: Training loss and testing loss for models.

## D  LOGICAL MODEL ANALYSIS

The fidelity and conciseness in 2.1 requirements enable us to construct a concise logical model with only a few salient AND-OR interactions as below:

$$\forall \mathbf{x}' \in \mathcal{Q}, \quad v(\mathbf{x}') \approx d'(\mathbf{x}'), \tag{14}$$

$$d'(\mathbf{x}') = \sum_{T \in \Omega_{\text{and}}^{\text{Salient}}} I_T^{\text{and}} \cdot \delta_{\text{and}} \left( \begin{smallmatrix} \mathbf{x}' \text{ triggers AND relation} \\ \text{between input variables in } T \end{smallmatrix} \right) + \sum_{T \in \Omega_{\text{or}}^{\text{Salient}}} I_T^{\text{or}} \cdot \delta_{\text{or}} \left( \begin{smallmatrix} \mathbf{x}' \text{ triggers OR relation} \\ \text{between input variables in } T \end{smallmatrix} \right) + b. \tag{15}$$

## E  EXPERIMENTAL DETAILS

### E.1  ARCHITECTURES AND DATASETS

In our comprehensive experimental framework, we employed a diverse range of deep learning architectures across multiple benchmark datasets to systematically analyze the strength of generalizable and non-generalizable interactions across varying interaction orders. For computer vision tasks, we utilized several convolutional neural network architectures: VGG-19 and ResNet-101/Resnet-50 were trained on the Tiny-ImageNet dataset, which contains 200 classes with 500 training images per class, downsampled to 64×64 pixels. Additionally, we trained VGG-16 on the CUB-200-2011 dataset (Caltech-UCSD Birds), which consists of 11,788 images across 200 bird species. For natural language processing tasks, we leveraged transformer-based models, specifically BERT-large (24 layers, 1024 hidden dimensions) and BERT-medium (8 layers, 512 hidden dimensions), which were trained on the Stanford Sentiment Treebank (SST-2) dataset containing binary sentiment classifications of movie reviews. Furthermore, we extended our NLP experiments by training both BERT-large and BERT-medium variants on the AG News dataset, which comprises news articles categorized into four classes.

For the training image preprocessing, the CUB-200-2011 dataset underwent bounding box processing to focus on the bird regions within each image, eliminating background noise that could potentially affect classification performance. The training images were then processed using data augmentation techniques including random resized cropping to 224×224 pixels, random horizontal flipping, tensor conversion, and normalization using ImageNet statistics. Similarly, Tiny-ImageNet images were preprocessed using data augmentation techniques, ensuring consistent input dimensions across all vision models despite the original differences in image resolutions between datasets.

To facilitate the training of both an original DNN and a corresponding baseline DNN, we randomly sampled and split the original training datasets to create two balanced subsets with equivalent data distributions. These equally distributed training subsets were used separately to train the original and baseline DNNs with different initialized parameters, maintaining a controlled experimental environment and ensuring that any observed differences between models could be attributed to our specific experimental manipulations rather than to variations in training data distribution.

### E.2  TRAINING LOSS AND TESTING LOSS

We report the training and testing loss for the model used in Section 3.3 as shown in Figure 6.

# F  AND-OR GRAPH

In this section we provide additional AND-OR logical models to explain the outputs of DeepSeek-R1-Distill-Llama-8B and Qwen2.5-7B across different prompts . The results detailed in Figure 7 show that the AND-OR logical model can explain LLMs' outputs on any randomly occluded samples.

**DeepSeek-Sample1**

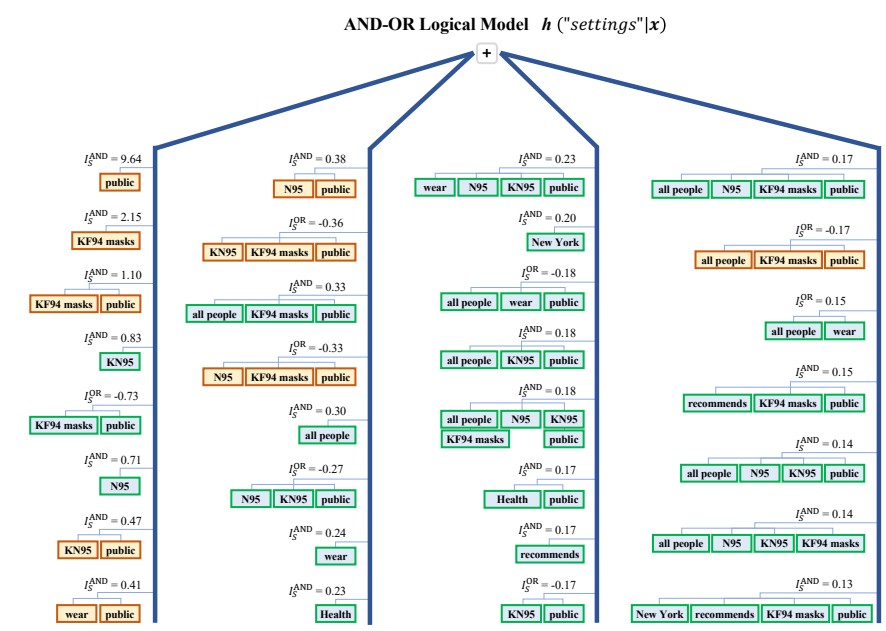

**Qwen-Sample1**

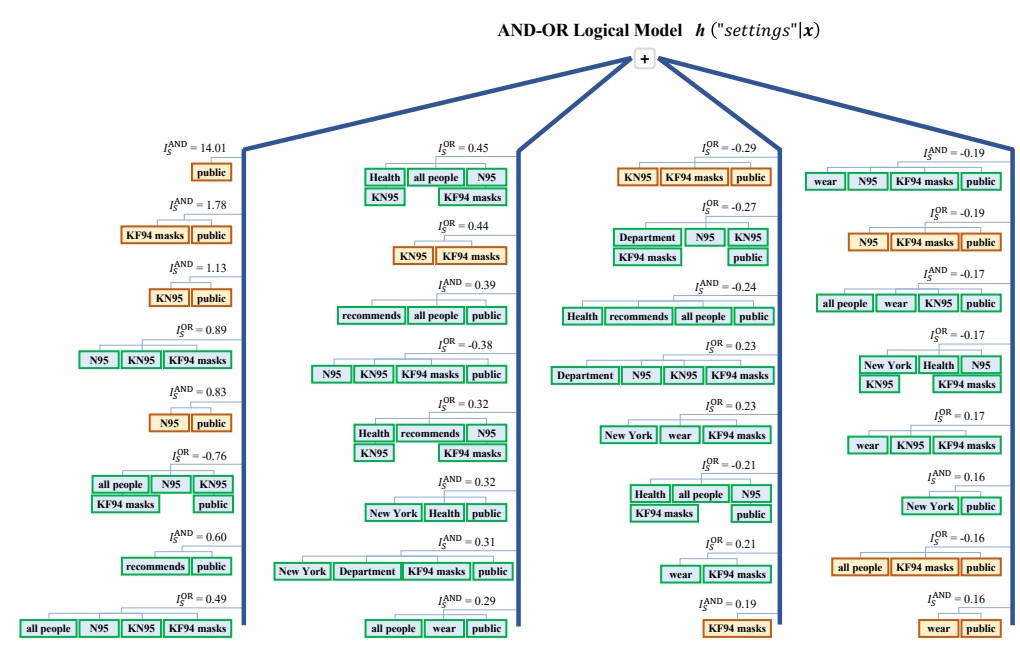

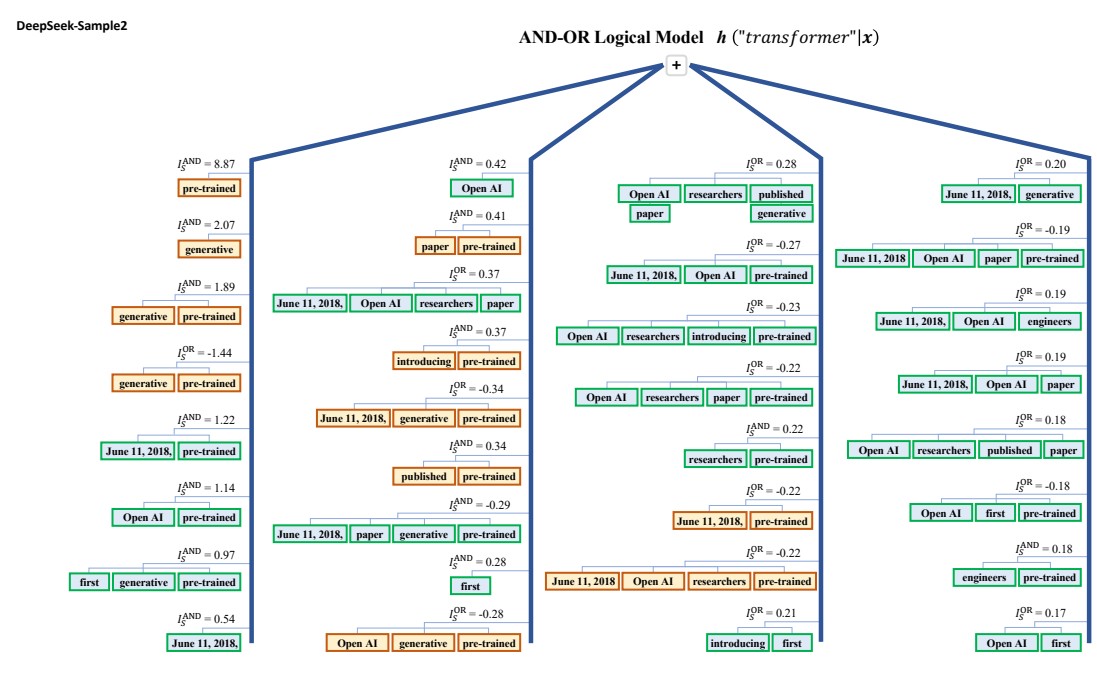

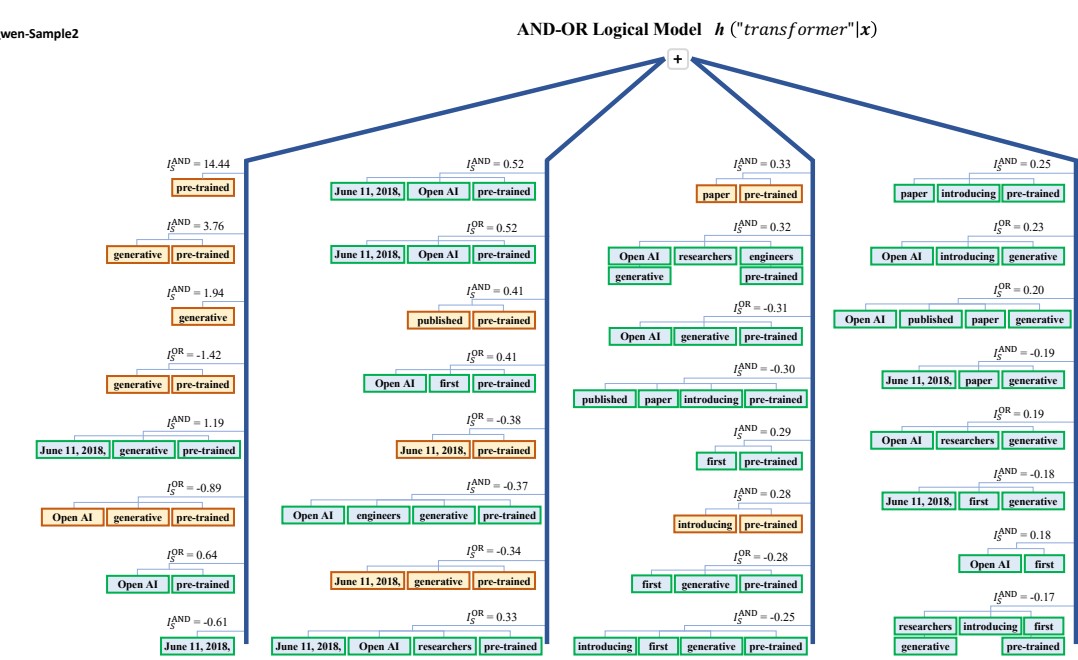

Figure 7: The AND-OR logical model successfully explains model outputs on arbitrarily occluded samples. The used distinct prompt-target pairs are : (1) Sample 1: "New York Department of Health recommends that all people should wear N95, KN95, or KF94 masks in all public" → "settings". Input variables: "New York", "Health", "recommends", "all people", "wear", "N95", "KN95", "KF94 masks", "public", "Health"; (2) Sample 2: "On June 11, 2018, OpenAI researchers and engineers published a paper introducing the first generative pre-trained" → "transformer". Input variables: "June 11, 2018,", "OpenAI", "researchers", "engineers", "published", "paper", "introducing", "first", "generative", "pre-trained".

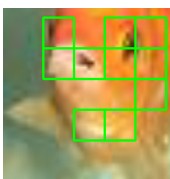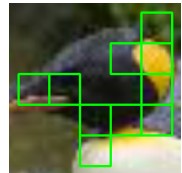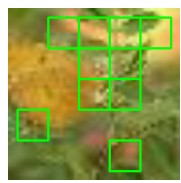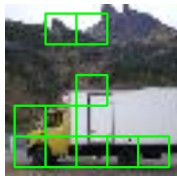

Figure 8: Examples of input variable selection on image data.

## G  EXAMPLES OF INPUT VARIABLES SELECTION

We provide examples of input variable selection on image data in Figure 8, where the input variables are highlighted with green boxes. And we provide examples of input variable selection on natural language data in Figure 9.

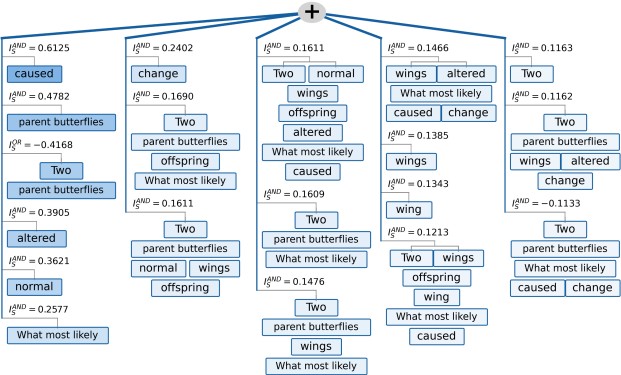

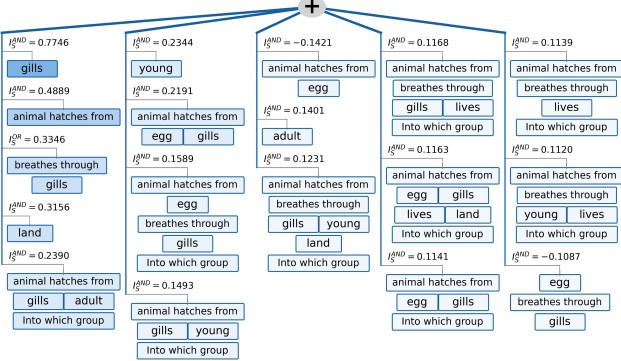

Figure 9: Examples of input variable selection on nlp data.

## H   PROOF OF EQUATION (11)

We want to derive Equation (11):

$$v(\mathbf{x}) = \sum_{\text{type}\in\{\text{and, or}\}} \sum_{S\in\Omega_{\text{type}}^{\text{Salient}}} I_S^{\text{type}} \cdot G_S^{\text{type}} \cdot \delta_{\text{type}} \left( \begin{smallmatrix} \mathbf{x}' \text{ triggers type relation} \\ \text{between input variables in } S \end{smallmatrix} \right)$$

$$+ \sum_{\substack{\text{type}\in \\ \{\text{and, or}\}}} \sum_{S\in\Omega_{\text{type}}^{\text{Salient}}} I_S^{\text{type}} \cdot (1 - G_S^{\text{type}}) \cdot \delta_{\text{type}} \left( \begin{smallmatrix} \mathbf{x}' \text{ triggers type relation} \\ \text{between input variables in } S \end{smallmatrix} \right)$$

$$+ \sum_{\text{type}\in\{\text{and, or}\}} \sum_{S\in\Omega_{\text{type}}^{\text{Non-Salient}}} I_S^{\text{type}} \cdot \delta_{\text{type}} \left( \begin{smallmatrix} \mathbf{x}' \text{ triggers type relation} \\ \text{between input variables in } S \end{smallmatrix} \right) + b$$

Let $v(\mathbf{x}')$ be the output of the DNN for a masked input $\mathbf{x}'$. From Theorem 1 (universal matching property), we know that the logical model $d(\mathbf{x}')$ precisely captures the DNN's output, so $v(\mathbf{x}') = d(\mathbf{x}')$. Equation (3) states:

$$d(\mathbf{x}') = \sum_{T\in\Omega_{\text{and}}} I_T^{\text{and}} \cdot \delta_{\text{and}} \left( \begin{smallmatrix} \mathbf{x}' \text{ triggers AND relation} \\ \text{between input variables in } T \end{smallmatrix} \right) + \sum_{T\in\Omega_{\text{or}}} I_T^{\text{or}} \cdot \delta_{\text{or}} \left( \begin{smallmatrix} \mathbf{x}' \text{ triggers OR relation} \\ \text{between input variables in } T \end{smallmatrix} \right) + b. \quad (16)$$

For consistency with Equation (11)'s summation index $S$, we rewrite Equation (16) using $S$ as the index:

$$v(\mathbf{x}') = \sum_{S\in\Omega_{\text{and}}} I_S^{\text{and}} \cdot \delta_S^{\text{and}}(\mathbf{x}') + \sum_{S\in\Omega_{\text{or}}} I_S^{\text{or}} \cdot \delta_S^{\text{or}}(\mathbf{x}') + b, \quad (17)$$

where $\delta_S^{\text{type}}(\mathbf{x}')$ is a shorthand for $\delta_{\text{type}} \left( \begin{smallmatrix} \mathbf{x}' \text{ triggers type relation} \\ \text{between input variables in } S \end{smallmatrix} \right)$. The term $v(\mathbf{x})$ on the LHS of Equation (11) can be interpreted as $v(\mathbf{x}')$ where $\mathbf{x}'$ is the specific input (potentially unmasked $\mathbf{x}$) for which the score is being decomposed. For generality, we proceed with $v(\mathbf{x}')$.

Equation (17) can be written more compactly as:

$$v(\mathbf{x}') = \sum_{\text{type}\in\{\text{and, or}\}} \sum_{S\in\Omega_{\text{type}}} I_S^{\text{type}} \cdot \delta_S^{\text{type}}(\mathbf{x}') + b. \quad (18)$$

The set of all interactions for a given type, $\Omega_{\text{type}}$, can be partitioned into salient interactions $\Omega_{\text{type}}^{\text{Salient}}$ and non-salient interactions $\Omega_{\text{type}}^{\text{Non-Salient}}$. These are defined as:

- $\Omega_{\text{type}}^{\text{Salient}} = \{S \mid |I_S^{\text{type}}| > \tau\}$
- $\Omega_{\text{type}}^{\text{Non-Salient}} = \{S \mid |I_S^{\text{type}}| \leq \tau\}$

Thus, $\Omega_{\text{type}} = \Omega_{\text{type}}^{\text{Salient}} \cup \Omega_{\text{type}}^{\text{Non-Salient}}$, and these two sets are disjoint.

Substituting this partition into the sum:

$$v(\mathbf{x}') = \sum_{\text{type}\in\{\text{and, or}\}} \left( \sum_{S\in\Omega_{\text{type}}^{\text{Salient}}} I_S^{\text{type}} \cdot \delta_S^{\text{type}}(\mathbf{x}') + \sum_{S\in\Omega_{\text{type}}^{\text{Non-Salient}}} I_S^{\text{type}} \cdot \delta_S^{\text{type}}(\mathbf{x}') \right) + b$$

$$= \sum_{\text{type}\in\{\text{and, or}\}} \sum_{S\in\Omega_{\text{type}}^{\text{Salient}}} I_S^{\text{type}} \cdot \delta_S^{\text{type}}(\mathbf{x}')$$

$$+ \sum_{\text{type}\in\{\text{and, or}\}} \sum_{S\in\Omega_{\text{type}}^{\text{Non-Salient}}} I_S^{\text{type}} \cdot \delta_S^{\text{type}}(\mathbf{x}') + b.$$

The second term here matches the "negligible non-salient interactions" part of Equation (11): $\sum_{\text{type}\in\{\text{and, or}\}} \sum_{S\in\Omega_{\text{type}}^{\text{Non-Salient}}} I_S^{\text{type}} \cdot \delta_S^{\text{type}}(\mathbf{x}')$.

Now, let's focus on the first term, which sums over salient interactions: $T_{\text{salient}} = \sum_{\text{type}\in\{\text{and, or}\}} \sum_{S\in\Omega_{\text{type}}^{\text{Salient}}} I_S^{\text{type}} \cdot \delta_S^{\text{type}}(\mathbf{x}')$. We introduce the generalizability indicator $G_S^{\text{type}}$ (defined

as Equation (5) to indicate whether a interaction is generalizable), where $G_S^{\text{type}} \in \{0, 1\}$. For any term $A_S^{\text{type}} = I_S^{\text{type}} \cdot \delta_S^{\text{type}}(\mathbf{x}')$ in the sum, we can use the identity $A_S^{\text{type}} = A_S^{\text{type}} \cdot G_S^{\text{type}} + A_S^{\text{type}} \cdot (1 - G_S^{\text{type}})$. So, $I_S^{\text{type}} \cdot \delta_S^{\text{type}}(\mathbf{x}') = I_S^{\text{type}} \cdot G_S^{\text{type}} \cdot \delta_S^{\text{type}}(\mathbf{x}') + I_S^{\text{type}} \cdot (1 - G_S^{\text{type}}) \cdot \delta_S^{\text{type}}(\mathbf{x}')$. Substituting this back into the sum $T_{\text{salient}}$:

$$
\begin{aligned}
T_{\text{salient}} &= \sum_{\text{type}\in\{\text{and, or}\}} \sum_{S\in\Omega_{\text{type}}^{\text{Salient}}} \left( I_S^{\text{type}} \cdot G_S^{\text{type}} \cdot \delta_S^{\text{type}}(\mathbf{x}') + I_S^{\text{type}} \cdot (1 - G_S^{\text{type}}) \cdot \delta_S^{\text{type}}(\mathbf{x}') \right) \\
&= \sum_{\text{type}\in\{\text{and, or}\}} \sum_{S\in\Omega_{\text{type}}^{\text{Salient}}} I_S^{\text{type}} \cdot G_S^{\text{type}} \cdot \delta_S^{\text{type}}(\mathbf{x}') \\
&\quad + \sum_{\text{type}\in\{\text{and, or}\}} \sum_{S\in\Omega_{\text{type}}^{\text{Salient}}} I_S^{\text{type}} \cdot (1 - G_S^{\text{type}}) \cdot \delta_S^{\text{type}}(\mathbf{x}').
\end{aligned}
$$

The first part of this expression corresponds to the "generalizable interaction" and the second part corresponds to the "non-generalizable interactions" in Equation (11).

Combining all components, we obtain:

$$
\begin{aligned}
v(\mathbf{x}') &= \sum_{\text{type}\in\{\text{and, or}\}} \sum_{S\in\Omega_{\text{type}}^{\text{Salient}}} I_S^{\text{type}} \cdot G_S^{\text{type}} \cdot \delta_S^{\text{type}}(\mathbf{x}') \\
&\quad + \sum_{\substack{\text{type}\in \\ \{\text{and, or}\}}} \sum_{S\in\Omega_{\text{type}}^{\text{Salient}}} I_S^{\text{type}} \cdot (1 - G_S^{\text{type}}) \cdot \delta_S^{\text{type}}(\mathbf{x}') \\
&\quad + \sum_{\text{type}\in\{\text{and, or}\}} \sum_{S\in\Omega_{\text{type}}^{\text{Non-Salient}}} I_S^{\text{type}} \cdot \delta_S^{\text{type}}(\mathbf{x}') + b.
\end{aligned}
$$

This matches equation 11. If Equation (11) is intended for the unmasked input $\mathbf{x}$, we simply set $\mathbf{x}' = \mathbf{x}$ throughout this derivation, and the LHS $v(\mathbf{x})$ becomes consistent.

## I   DEFINE THE THREE-PHASE MORE QUANTITATIVELY

Adopting a more rigorous quantitative approach helps to better delineate the boundaries between the three phases. Therefore, we have formulated more precise, quantitative definitions for these phases below.

We characterize the learning dynamics using several key metrics. Let $\delta_g(t)$ and $\delta_n(t)$ be the incremental strengths of new **generalizable** and **non-generalizable** interactions learned at time $t$. We also track the average interaction generalization power, $\overline{\mathcal{H}}(t)$, and the number of learned interactions, $\overline{\mathcal{N}}(t)$.

We can now characterize the three phases:

- **Phase 1 (Interval: $[t_0, t_1]$)**
  The model primarily learns generalizable patterns. $\overline{\mathcal{H}}(t)$ increases while $\overline{\mathcal{N}}(t)$ decreases. This phase concludes when $\overline{\mathcal{H}}(t)$ reaches its peak:
  $$t_1 = \arg\max_t \overline{\mathcal{H}}(t).$$

- **Phase 2 (Interval: $(t_1, t_2]$)**
  The model begins to learn non-generalizable interactions along with generalizable interactions. $\overline{\mathcal{H}}(t)$ decreases and $\overline{\mathcal{N}}(t)$ increases. Learning of generalizable patterns still dominates, satisfying the condition
  $$\delta_g(t) \geq r \cdot \delta_n(t).$$
  This phase ends at time $t_2$, the first point where the strengths balance:
  $$\delta_g(t_2) = r \cdot \delta_n(t_2),$$
  given the monotonic increase of the ratio $\dfrac{\Delta_n(t)}{\Delta_g(t)}$.

- **Phase 3 (Interval:** $(t_2, t_3]$**)**
  The acquisition of new interactions is now dominated by non-generalizable ones, with their strengths satisfying
  $$\delta_g(t) < r \cdot \delta_n(t).$$
  Moreover, $\overline{\mathcal{N}}(t)$ continues to increase and $\overline{\mathcal{H}}(t)$ continues to decrease.

## J  EXPERIMENT OF REMOVING THE NON-GENERALIZABLE INTERACTIONS FROM THE MODEL OUTPUT

### J.1  SAMPLE SELECTION FOR LOSS COMPUTATION

The loss compare in Section 3.2, encompassing both standard and modified loss calculations after removing non-generalizable interactions, were derived from an average over 100 samples per dataset. For the Tiny-ImageNet dataset, these 100 samples were obtained by first randomly selecting 10 classes, and then randomly drawing 10 image instances from each chosen class. This sample selection process was applied independently to both the training and testing sets to gather the respective samples for loss calculation. For SST-2 and AG News datasets, 100 samples were randomly drawn from within all available classes, aiming for a proportional representation or a balanced distribution across classes (e.g., attempting to sample $m = 100/C$ instances from each of the $C$ classes, with necessary adjustments made for class imbalances or varying class sizes).

### J.2  FUTURE WORK AND POTENTIAL DIRECTIONS

The demonstration that non-generalizable interactions are a key driver of overfitting, as discussed in Section 3.2, suggests several compelling research avenues. Future efforts could focus on developing novel training paradigms—encompassing targeted regularization, interaction-aware network pruning, or refined early stopping criteria—all designed to proactively prevent or penalize the formation of these detrimental interactions. Another promising direction involves moving beyond simple removal, exploring methods to dynamically modulate or down-weight the influence of identified non-generalizable interactions during training or inference, potentially guided by a deeper understanding of their emergence.

## K  EXPERIMENT OF PENALIZING NON-GENERALIZABLE INTERACTIONS IN DNN

To bridge the theoretical discussion in our paper with practical implementation, we conducted a follow-up experiment. The core idea was to leverage a dedicated validation set to identify and subsequently penalize non-generalizable interactions and classification loss during the end-to-end training process.

Our approach consisted of two key steps. First, to **quantify the generalization power of interactions**, we trained a baseline Deep Neural Network (DNN) on a separate validation set. This model allowed us to quantify which of the learned feature interactions were non-generalizable. Second, to **design the penalty term**, we designed a specialized loss function based on the quantification from the previous step. During the main model's training, this function directly penalized the absolute values of the interactions identified as non-generalizable, denoted as $|I_S^{\text{NoneG}}|$, along with minimization of the classification loss.

We evaluated this approach on the CIFAR-10 dataset using a VGG-11 architecture, and the results were encouraging. The method led to a clear improvement in the quality of learned interactions: The proportion of generalizable interactions, as measured by $\overline{\mathcal{H}}$ defined in Section 3.1, increased significantly from 28% to 39%. Correspondingly, the proportion of non-generalizable interactions decreased from 72% to 61%.

**Regarding the impact on model accuracy**, the proposed method consistently improves performance across different architectures and datasets. For the VGG-16 model on the Tiny-ImageNet dataset, test accuracy increased by **2.5% (46.3% to 48.8%)**. Similarly, for the BERT-Tiny model on the SST-2 sentiment classification task, accuracy improved by **4.4% (72.2% to 76.6%)**. The algorithm also

demonstrates significant gains in few-shot learning scenarios. For instance, in an object recognition task with only 200 training samples, our method achieved a **6.0% (76.6% to 82.6%)** improvement in classification accuracy using a ResNet-18 model.

In summary, this experiment demonstrated that our proposed method effectively reduced the learning of non-generalizable interactions and improved the neural network's performance, showing particular promise in situations with limited training data.

## L    COMPUTATIONAL RESOURCES AND PROCEDURAL DURATIONS

All experiments were conducted on a dedicated server equipped with four NVIDIA GeForce RTX 3080 Ti GPUs and powered by Intel(R) Xeon(R) Gold 6146 CPU.

The model training durations varied depending on the specific experiment and model complexity:

- **Ablation study:** The training for ablation studies, involving up to five baseline Deep Neural Networks (DNNs), represented the most time-consuming part. Due to the necessity of training these models sequentially or on separate GPUs to ensure resource isolation, the cumulative training time for these experiments was approximately 50 hours.
- **Small to medium models:** For other experiments involving small to medium-sized models such as ResNet-50, individual training runs were typically completed within 12 hours per model.
- **Larger models:** Training larger models, including ResNet-101 and BERT-large, for the remaining experimental setups generally concluded within 30 hours per model.

Regarding specific experimental procedures:

- **Interaction extraction:** The process of extracting each individual interaction took approximately 60 seconds.
- **Generalizable interaction assessment:** The evaluation of interaction generalization was comparatively swift. Assessing the generalizable power for a set of 100 samples typically completed within a range of 8 to 14 seconds.

These resources and timeframes allowed for comprehensive experimentation and validation of the proposed methods.

## M    REDEFINITION OF THE GENERALIZATION POWER METRIC

In this section, we provide the redefinition of the generalizability metric for interactions in 5. Instead of the binary metrics, we use a continuous metric within the range $[0, 1]$ to quantify the generalization power of interactions between the target model $v$ and the baseline model $v^{\text{base}}$.

For each salient interaction $S$ (an AND interaction or an OR interaction), the generalization power is redefined as the similarity between the interaction effect in the target model and that in the baseline model as follows:

$$\mathcal{G}'^{\text{and}}_{S,v^{\text{base}}} = \mathbf{1}\left(I^{\text{and}}_S \cdot I^{\text{and}}_{S,v^{\text{base}}} > 0\right) \cdot \frac{\min\left(\left|I^{\text{and}}_S\right|, \left|I^{\text{and}}_{S,v^{\text{base}}}\right|\right)}{\max\left(\left|I^{\text{and}}_S\right|, \left|I^{\text{and}}_{S,v^{\text{base}}}\right|\right)}, \tag{19}$$

$$\mathcal{G}'^{\text{or}}_{S,v^{\text{base}}} = \mathbf{1}\left(I^{\text{or}}_S \cdot I^{\text{or}}_{S,v^{\text{base}}} > 0\right) \cdot \frac{\min\left(\left|I^{\text{or}}_S\right|, \left|I^{\text{or}}_{S,v^{\text{base}}}\right|\right)}{\max\left(\left|I^{\text{or}}_S\right|, \left|I^{\text{or}}_{S,v^{\text{base}}}\right|\right)}. \tag{20}$$

Here, $\mathbf{1}(\cdot)$ is an indicator function, which returns 1 if the condition holds, and returns 0 otherwise. The ratio term $\frac{\min(|I_S|, |I_{S,v^{\text{base}}}|)}{\max(|I_S|, |I_{S,v^{\text{base}}}|)} \in [0, 1]$ measures the similarity of the interaction effects encoded by

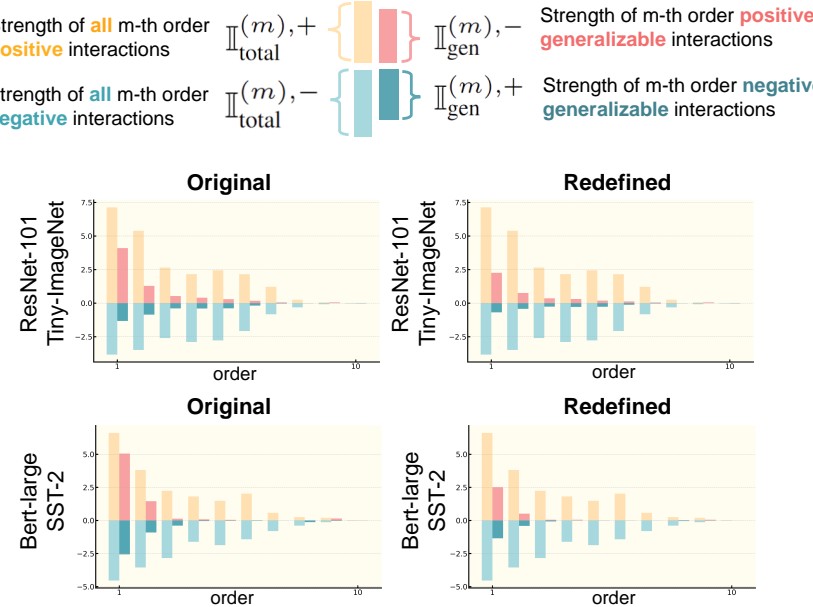

Figure 10: Analysis of generalization power across interaction orders. (Left) The distribution of interactions calculated with the original metric ($\mathcal{G}$) titled with "**Origin**". (Right) The distribution The distribution of interactions calculated with the redefined metric ($\mathcal{G}'$) titled with "**Redefined**".

the two models, which reflects generalizability of this interaction. Only when the same interaction $S$ (or OR interaction $S$) is simultaneously extracted by both DNNs and exhibits exactly the same effect, we consider it to be 100.0% generalizable.

**Normalization.** Considering that the outputs of different DNNs often exhibit different ranges of perturbation, it is necessary to normalize the interaction effects to ensure fair comparison. We normalize $I_S^{\text{and}}$ and $I_S^{\text{or}}$ as: $I_S^{\text{and}} \leftarrow \frac{I_S^{\text{and}}}{\mathbb{E}_{\mathbf{x}}[|v(N)-v(\emptyset)|]}$, and $I_S^{\text{or}} \leftarrow \frac{I_S^{\text{or}}}{\mathbb{E}_{\mathbf{x}}[|v(N)-v(\emptyset)|]}$, where $\mathbb{E}_{\mathbf{x}}[|v(N)-v(\emptyset)|]$ represents the expected output variation of the model.

Using this redefined metric $\mathcal{G}'^{\text{and}}_{S,v^{\text{base}}}$ and $\mathcal{G}'^{\text{or}}_{S,v^{\text{base}}}$, we analyzed the distribution of generalizable interactions across different interaction orders on ResNet-101 trained with Tiny-ImageNet and Bert-large trained with SST-2. As shown in Figure 10, the new results aligned closely with results based on the old metric. Because the new metric is more stringent, the strength of generalizable interactions is reduced. However, the distribution of generalizable interactions over different orders based on the new metric is still similar to that based on the old metric.

## N    EVALUATION OF THE QUANTIFICATION OF GENERALIZATION POWER

We conducted a new experiment to explicitly evaluate the quantification of generalization power. Specifically, we uniformly partitioned all data into three sets (A, B, and C) to perform cross-validation. We trained the target DNN on the dataset A, and trained two baseline DNNs on the dataset B and the dataset C, respectively. Here we regarded the baseline model trained on dataset B as representing the interactions on the validation set and regarded the baseline model trained on dataset C as representing the interactions on the test set. Then, given each input sample, we followed Equation (7) in Section 2.2 in the manuscript to measure the distribution (i.e., $\mathbb{I}_{\text{gen}_B}^{(m),+}$ and $\mathbb{I}_{\text{gen}_B}^{(m),-}$) of interactions that could generalize to the validation set (i.e., the set B) and the distribution (i.e., $\mathbb{I}_{\text{gen}_C}^{(m),+}$ and $\mathbb{I}_{\text{gen}_C}^{(m),-}$) of interactions that can generalize to the test set (i.e., the set C).[11]

---

[11]Specifically, extending the definition in Equation (7), we utilized $\mathbb{I}_{\text{gen}_B}^{(m),+}$ to denote the aggregate effect strength of all generalizable salient interactions with positive effect extracted from the baseline DNN trained on

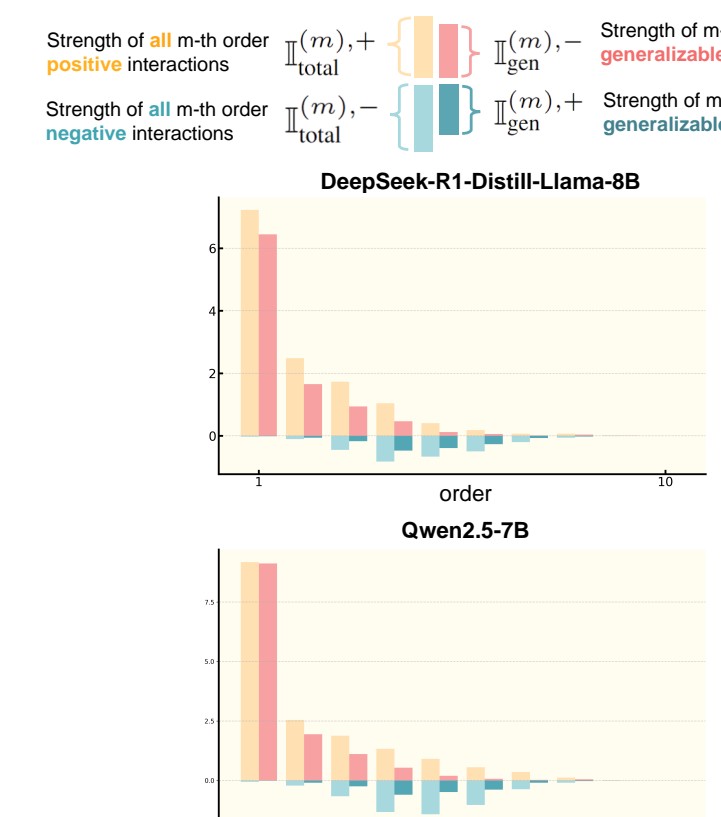

Figure 11: Distribution of interactions measured between Qwen2.5-7B (baseline) and DeepSeek-R1-Distill-Llama-8B (target) using SQuAD data following Equation 7 in Section 2.2. The plot illustrates that the two LLMs tend to encode low-order, simple interactions that are more transferable.

Table 1: Jaccard Similarity ($J$) of generalizable interaction distributions between validation set (B) and test set (C).

| Model and Dataset | Jaccard Similarity ($J$) |
|---|---|
| ResNet-50 on Tiny-ImageNet | 71.4% |
| Bert-large on SST-2 | 74.7% |

The validation was conducted to test whether the interactions, which could generalize to the validation set, could also generalizable to the test set. If feasible, the faithfulness of our method was validated. Specifically, we computed the Jaccard similarity between the two distributions as follows:

$$
J = \frac{\sum\limits_{m=1}^{n} \left[ \min\left( \left| \mathbb{I}_{\mathrm{gen_B}}^{(m),+} \right|, \left| \mathbb{I}_{\mathrm{gen_C}}^{(m),+} \right| \right) + \min\left( \left| \mathbb{I}_{\mathrm{gen_B}}^{(m),-} \right|, \left| \mathbb{I}_{\mathrm{gen_C}}^{(m),-} \right| \right) \right]}{\sum\limits_{m=1}^{n} \left[ \max\left( \left| \mathbb{I}_{\mathrm{gen_B}}^{(m),+} \right|, \left| \mathbb{I}_{\mathrm{gen_C}}^{(m),+} \right| \right) + \max\left( \left| \mathbb{I}_{\mathrm{gen_B}}^{(m),-} \right|, \left| \mathbb{I}_{\mathrm{gen_C}}^{(m),-} \right| \right) \right]}.
$$

dataset B, $\mathbb{I}_{\mathrm{gen_B}}^{(m),-}$ to denote the aggregate effect strength of all generalizable salient interactions with negative effect extracted from the baseline DNN trained on dataset B, $\mathbb{I}_{\mathrm{gen_C}}^{(m),+}$ to denote the aggregate effect strength of all generalizable salient interactions with positive effect extracted from the baseline DNN trained on dataset C and $\mathbb{I}_{\mathrm{gen_C}}^{(m),-}$ to denote the aggregate effect strength of all generalizable salient interactions with negative effect extracted from the baseline DNN trained on dataset C.

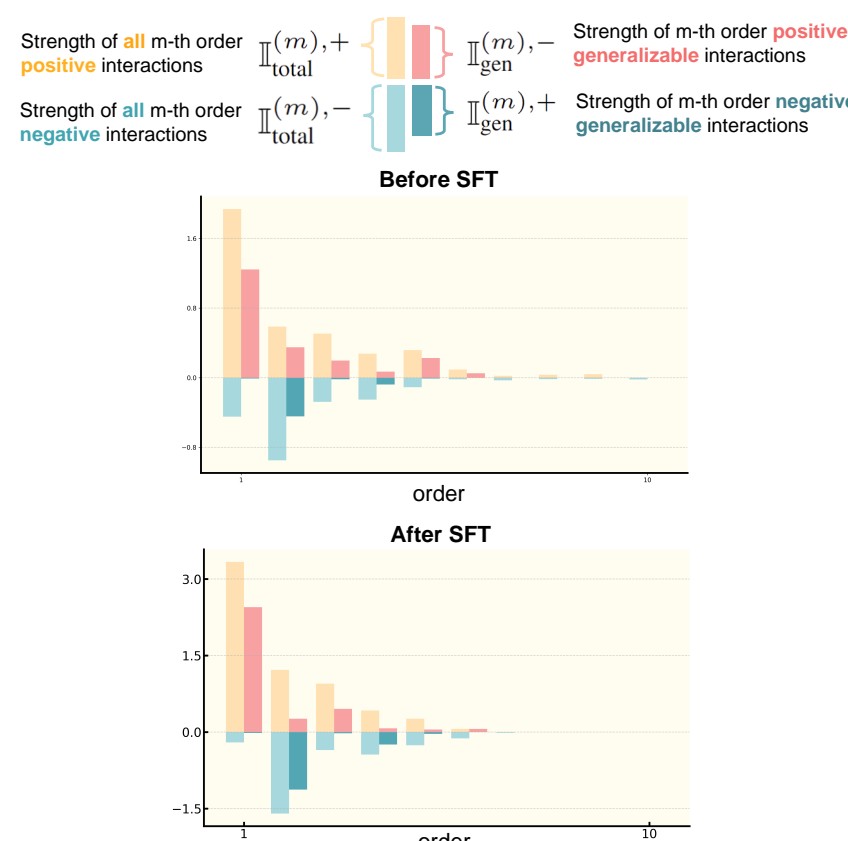

Figure 12: Distribution of interactions before and after the SFT process.

The experimental results in Table 1 showed that the DNNs exhibited high Jaccard similarity of generalizable interaction distribution between the test set and the validation set. This supported the faithfulness of our method of identifying generalizable interactions.

## O    EXPERIMENT ON TRANSFORMER-BASED LLMS

Extensively, we have conducted an experiment of analyzing the distribution of generalizable interactions on several open source LLMs based on the transformer architecture, including Qwen2.5-7B and DeepSeek-R1-Distill-Llama-8B, which are well-known large language models and have been widely applied.

Notably, we acknowledge the significant challenge of retraining a foundational model from scratch to serve as a baseline. Therefore, considering that different LLMs released by various organizations/companies are typically trained on different and massive datasets, we provisionally utilized one LLM as the baseline DNN to evaluate another.

Specifically, we selected Qwen2.5-7B model as the baseline model and DeepSeek-R1-Distill-Llama-8B model as the target model. We used data from the SQuAD dataset and followed Equation (7) from Section 2.2 in the manuscript to measure the distribution of interactions that can generalize to the baseline DNN.

As illustrated in Figure 11, the new experimental results suggested that the two large models tended to encode low-order, simple generalizable interactions. This indicated that these well-trained LLMs might focus on learning simple patterns during training, and such simple interactions could potentially be more transferable between models. A systematic investigation of transferability across a larger set of LLMs and datasets is left to future work.

## P    ANALYZE LLM FINE-TUNING WITH OUR METHOD

For the SFT process, we used the Unilaw-R1-Data (Cai et al., 2025) to fine-tune the Qwen2.5-7B-Instruct model, which was originally trained on 18T tokens of data. Experimental results showed that during the SFT process, low-order interactions increased, while high-order interactions decreased. The proportion of generalizable interactions changed from 45.2% to 52.3%. The current experimental results had shown that the SFT process was still within the first phase. Figure 12 shows the distribution of interactions before and after the SFT process. We will continue to finetune the LLM and keep reporting latest distribution of interactions to provide dynamics in the second and third phases in future. The above experiments show that our method remains applicable to analyzing LLM training.

## Q    THE USE OF LARGE LANGUAGE MODELS (LLMS)

In the preparation of this article, Large Language Models (LLMs) were utilized solely for the purpose of language polishing and identifying grammatical inaccuracies. The core research, analysis, findings, and intellectual contributions remain entirely those of the authors. The LLMs acted as an assistive tool to improve the clarity and fluency of the academic prose, without influencing the substantive content or the logical arguments presented.

