# OpenReview forum: "Towards the Three-Phase Dynamics of Generalization Power of a DNN"
_ICLR.cc/2026/Conference — Submitted to ICLR 2026_

### Official Review · Reviewer_5ny7 · 2025-10-23

**Soundness:** 2
**Presentation:** 3
**Contribution:** 2
**Rating:** 4
**Confidence:** 2

**Summary:**

The paper proposes measuring DNN’s generalization power by decomposing its predictions into AND–OR interactions. It also proposes a three-phase dynamics of the generalization power of interactions.

**Strengths:**

The proposed three-stage dynamics sounds interesting and reasonable. The experiments results span multiple architectures and domains.

**Weaknesses:**

The non-generalizable interactions can only be identified by training the model on the test data, which is not realistic.

**Questions:**

Where did you define non-generalizable interactions? How do you identify them? Are those interactions data-specific or they are invariant across different datasets?

Training on test data sounds weird. Is it possible to identify the non-generalizable interactions only using train and evaluation data, and the identified interactions can be transferred to test data?

Can the framework of identifying non-generalizable interactions be transferred to transformers?

---

> ### Author Response · Authors · 2025-11-21
> **Rebuttal by Authors part1**
>
> Thank you for your efforts on the review. We will try our best to answer all your questions.
>
> **If you are not satisfied with our answers or have more questions, please let us know as soon as possible.**
>
> ------
>
> ## Q1: The non-generalizable interactions can only be identified by training the model on the test data, which is not realistic. Training on test data sounds weird. Is it possible to identify the non-generalizable interactions only using train and evaluation data, and the identified interactions can be transferred to test data?
>
> > "The non-generalizable interactions can only be identified by training the model on the test data, which is not realistic."
> > "Training on test data sounds weird. Is it possible to identify the non-generalizable interactions only using train and evaluation data, and the identified interactions can be transferred to test data?"
>
> **A**: A good question. We uniformly partitioned all data into three sets (A, B, and C) to perform cross-validation. We trained the target DNN on the dataset A, and trained two baseline DNNs on the dataset B and the dataset C, respectively. Here, we regarded the baseline model trained on dataset B as representing the interactions on the validation set and regarded the baseline model trained on dataset C as representing the interactions on the test set. Then, given each input sample, we followed Equation (7) in Section 2.2 to measure the distribution of interactions that could generalize to the validation set (the set B), i.e., \\(\\mathbb{I}\_\{\\text{gen}\_\\text{B}\}^{(m),+}\\) and \\(\\mathbb{I}\_\{\\text{gen}\_\\text{B}\}^{(m),-}\\). We also measured the distribution of interactions that could generalize to the test set (the set C), i.e., \\(\\mathbb{I}\_\{\\text{gen}\_\\text{C}\}^{(m),+}\\) and \\(\\mathbb{I}\_\{\\text{gen}\_\\text{C}\}^{(m),-}\\).[^1]
>
> The validation was conducted to test whether the interactions, which could generalize to the validation set, could also generalize to the test set. If feasible, the faithfulness of our method was validated. Specifically, we computed the Jaccard similarity between the two distributions as follows:
>
> \\[
> J = \\frac{ \\displaystyle \\sum\_\{m=1\}^{n} \\Big[ \\min\\!\\big(\\,\\left|\\mathbb{I}\_\{\\text{gen}\_\\text{B}\}^{(m),+}\\right|,\\,\\left|\\mathbb{I}\_\{\\text{gen}\_\\text{C}\}^{(m),+}\\right|\\big) + \\min\\!\\big(\\,\\left|\\mathbb{I}\_\{\\text{gen}\_\\text{B}\}^{(m),-}\\right|,\\,\\left|\\mathbb{I}\_\{\\text{gen}\_\\text{C}\}^{(m),-}\\right|\\big) \\Big] }{ \\displaystyle \\sum\_\{m=1\}^{n} \\Big[ \\max\\!\\big(\\,\\left|\\mathbb{I}\_\{\\text{gen}\_\\text{B}\}^{(m),+}\\right|,\\,\\left|\\mathbb{I}\_\{\\text{gen}\_\\text{C}\}^{(m),+}\\right|\\big) + \\max\\!\\big(\\,\\left|\\mathbb{I}\_\{\\text{gen}\_\\text{B}\}^{(m),-}\\right|,\\,\\left|\\mathbb{I}\_\{\\text{gen}\_\\text{C}\}^{(m),-}\\right|\\big) \\Big] }
> \\]
>
> The experimental results showed that the DNNs exhibited high Jaccard similarity of generalizable interaction distribution between the test set and the validation set. This supported the faithfulness of our method of identifying generalizable interactions.
>
> We have added the detailed experimental settings and results in the newly added Appendix N.
>
> ---
>
> [^1]: Specifically, extending the definition in Equation (7), we utilized \\(\\mathbb{I}\_\{\\text{gen}\_\\text{B}\}^{(m),+}\\) to denote the aggregate effect strength of all generalizable salient interactions with **positive** effect extracted from the baseline DNN trained on dataset B, and \\(\\mathbb{I}\_\{\\text{gen}\_\\text{B}\}^{(m),-}\\) to denote the aggregate effect strength of all generalizable salient interactions with **negative** effect extracted from the baseline DNN trained on dataset B. Similarly, \\(\\mathbb{I}\_\{\\text{gen}\_\\text{C}\}^{(m),+}\\) and \\(\\mathbb{I}\_\{\\text{gen}\_\\text{C}\}^{(m),-}\\) denote the aggregate effect strengths (positive and negative, respectively) of generalizable salient interactions extracted from the baseline DNN trained on dataset C.

---

> ### Author Response · Authors · 2025-11-21
> **Rebuttal by Authors part2**
>
> ## Q2: Where did you define non-generalizable interactions? How do you identify them?
>
> > "Where did you define non-generalizable interactions? How do you identify them?"
>
> **A**: We have defined non-generalizable interactions in Line 208–229, Section 2.2.1. The entire Section 2.2.1 in the manuscript defines a metric to quantify the generalizability of interactions. Specifically, we quantify the generalizability of individual interactions encoded by the target DNN to a baseline DNN, which is denoted by $v^{\\text{base}}$ and trained on the testing samples (please refer to Appendix E.1 for details). To elucidate this approach, let us consider an extracted salient AND interaction $S$, s.t. $|I^{\\text{and}}\_S| > \\tau$. To this end, the interaction $S$ is deemed to represent an inference pattern that can generalize to the testing set, if the baseline DNN $v^{\\text{base}}$ also extracts the same AND interaction $S$ with a salient effect (i.e., $|I^{\\text{and}}\_{S,v^{\\text{base}}}| > \\tau$) from the input, and the AND interaction $S$ exerts a consistent directional influence on the classification of $\\mathbf{x}$ (i.e., $I^{\\text{and}}\_{S,v^{\\text{base}}} \\cdot I^{\\text{and}}\_S > 0$). The same principle also applies to OR interactions. Accordingly, the generalizability of an AND/OR interaction is assessed by the following binary metrics:
>
> $$
> \\begin{aligned}
>     \\mathcal{G}^{\\text{and}}\_{S,v^{\\text{base}}} &= \\mathbf{1}\\big(\\lvert I^{\\text{and}}\_{S,v^{\\text{base}}}\\rvert > \\tau\\big)\\cdot \\mathbf{1}\\big(I^{\\text{and}}\_S \\cdot I^{\\text{and}}\_{S,v^{\\text{base}}} > 0\\big) \\in \\{0,1\\},\\\\[6pt]
>     \\mathcal{G}^{\\text{or}}\_{S,v^{\\text{base}}} &= \\mathbf{1}\\big(\\lvert I^{\\text{or}}\_{S,v^{\\text{base}}}\\rvert > \\tau\\big)\\cdot \\mathbf{1}\\big(I^{\\text{or}}\_S \\cdot I^{\\text{or}}\_{S,v^{\\text{base}}} > 0\\big) \\in \\{0,1\\},
> \\end{aligned}
> $$
>
> where $\\mathbf{1}(\\cdot)$ is a binary indicator function that outputs $1$ only when the condition is satisfied.
>
> Accordingly, when the metric $\\mathcal{G}^{\\text{and}}\_{S,v^{\\text{base}}}=0$ (or $\\mathcal{G}^{\\text{or}}\_{S,v^{\\text{base}}}=0$), the AND (OR) interaction $S$ is considered a non-generalizable interaction.
>
> Please see Equation (5), Line 208–229, Section 2.2.1 for detail.
>
> **We are not sure whether the above answer adequately addresses your concern. If you have any further questions, please feel free to let us know.**

---

> ### Author Response · Authors · 2025-11-21
> **Rebuttal by Authors part3**
>
> ## Q3: Are those interactions data-specific or they are invariant across different datasets?
> > "Are those interactions data-specific or they are invariant across different datasets?"
>
> **A:** A good question. Please see the **new experiment** in answers to Q1. We have followed your suggestions to conduct a new experiment to address your concern. To test whether the extracted generalizable interactions can indeed generalize to the test set, we uniformly partitioned all data into three sets (A, B, and C) to perform cross-validation.
>
> Specifically, We uniformly partitioned all data into three sets (A, B, and C) to perform cross-validation. We trained the target DNN on the dataset A, and trained two baseline DNNs on the dataset B and the dataset C, respectively. Here, we regarded the baseline model trained on dataset B as representing the interactions on the validation set and regarded the baseline model trained on dataset C as representing the interactions on the test set. Then, given each input sample, we followed Equation (7) in Section 2.2 to measure the distribution of interactions that could generalize to the validation set (the set B), i.e., \\(\\mathbb{I}\_\{\\text{gen}\_\\text{B}\}^{(m),+}\\) and \\(\\mathbb{I}\_\{\\text{gen}\_\\text{B}\}^{(m),-}\\). We also measured the distribution of interactions that could generalize to the test set (the set C), i.e., \\(\\mathbb{I}\_\{\\text{gen}\_\\text{C}\}^{(m),+}\\) and \\(\\mathbb{I}\_\{\\text{gen}\_\\text{C}\}^{(m),-}\\).[^1]
>
> The validation was conducted to test whether the interactions, which could generalize to the validation set, could also generalize to the test set. If feasible, the faithfulness of our method was validated. Specifically, we computed the Jaccard similarity between the two distributions as follows:
>
> \\[
> J = \\frac{ \\displaystyle \\sum\_\{m=1\}^{n} \\Big[ \\min\\!\\big(\\,\\left|\\mathbb{I}\_\{\\text{gen}\_\\text{B}\}^{(m),+}\\right|,\\,\\left|\\mathbb{I}\_\{\\text{gen}\_\\text{C}\}^{(m),+}\\right|\\big) + \\min\\!\\big(\\,\\left|\\mathbb{I}\_\{\\text{gen}\_\\text{B}\}^{(m),-}\\right|,\\,\\left|\\mathbb{I}\_\{\\text{gen}\_\\text{C}\}^{(m),-}\\right|\\big) \\Big] }{ \\displaystyle \\sum\_\{m=1\}^{n} \\Big[ \\max\\!\\big(\\,\\left|\\mathbb{I}\_\{\\text{gen}\_\\text{B}\}^{(m),+}\\right|,\\,\\left|\\mathbb{I}\_\{\\text{gen}\_\\text{C}\}^{(m),+}\\right|\\big) + \\max\\!\\big(\\,\\left|\\mathbb{I}\_\{\\text{gen}\_\\text{B}\}^{(m),-}\\right|,\\,\\left|\\mathbb{I}\_\{\\text{gen}\_\\text{C}\}^{(m),-}\\right|\\big) \\Big] }
> \\]
>
> The experimental results showed that the DNNs exhibited high Jaccard similarity of generalizable interaction distribution between the test set and the validation set. This supported the faithfulness of our method of identifying generalizable interactions.
>
> We have added the detailed experimental settings and results in the newly added Appendix N.
>
> -----
>
> [^1]: Specifically, extending the definition in Equation (7), we utilized \\(\\mathbb{I}\_\{\\text{gen}\_\\text{B}\}^{(m),+}\\) to denote the aggregate effect strength of all generalizable salient interactions with **positive** effect extracted from the baseline DNN trained on dataset B, and \\(\\mathbb{I}\_\{\\text{gen}\_\\text{B}\}^{(m),-}\\) to denote the aggregate effect strength of all generalizable salient interactions with **negative** effect extracted from the baseline DNN trained on dataset B. Similarly, \\(\\mathbb{I}\_\{\\text{gen}\_\\text{C}\}^{(m),+}\\) and \\(\\mathbb{I}\_\{\\text{gen}\_\\text{C}\}^{(m),-}\\) denote the aggregate effect strengths (positive and negative, respectively) of generalizable salient interactions extracted from the baseline DNN trained on dataset C.

---

> ### Author Response · Authors · 2025-11-21
> **Rebuttal by Authors part4**
>
> ## Q4: Can the framework of identifying non-generalizable interactions be transferred to transformers?
>
> >"Can the framework of identifying non-generalizable interactions be transferred to transformers?"
>
> **A:** A good question. In fact, our experiments in Figure 3 have already shown results on transformer-based DNNs. The BERT-medium model and the BERT-large model in Figure 3 are two typical transformer-based DNNs. The experimental results have shown that the distribution of interactions follows a three-phase dynamics during training process of the DNN.
>
> More importantly, we have also conducted a **new experiment** on several open source LLMs based on the transformer architecture, including Qwen2.5-7B, and DeepSeek-R1-Distill-Llama-8B, which were well-known large language models and have been widely applied.
>
> Notably, we acknowledged the significant challenge of retraining a foundational model from scratch to serve as a baseline. Therefore, considering that different LLMs released by various organizations/companies were typically trained on different and massive datasets, we provisionally utilized one LLM as the baseline DNN to evaluate another.
>
> Specifically, we selected Qwen2.5-7B model as the baseline model and DeepSeek-R1-Distill-Llama-8B model as the target model. We used data from the SQuAD dataset and followed Equation (7) from Section 2.2 in the manuscript to measure the distribution of interactions that could generalize to the baseline DNN.
>
> The new experimental results suggested that the two LLMs tended to encode low-order (simple) generalizable interactions. This indicated that these well-trained LLMs might focus on learning simple patterns during training. And such simple interactions could potentially be more transferable between models.
>
> Please see the newly added Appendix O for results.
>
> ------
>
> **Please let us know if you have any further concerns. We will respond promptly.**

---

> ### Author Response · Authors · 2025-11-25
> **Follow-Up on Rebuttal**
>
> Dear Reviewer 5ny7,
>
> We hope this message finds you well. Should you have any further questions or concerns, please do not hesitate to contact us. We will respond at your earliest convenience.
>
> Best regards,
>
> The authors

---

### Official Review · Reviewer_K1xr · 2025-10-29

**Soundness:** 4
**Presentation:** 2
**Contribution:** 4
**Rating:** 8
**Confidence:** 3

**Summary:**

Recent advances in symbolic generalization show that, given a DNN and an input, there exists an AND-OR logical model (a sum of AND and OR interactions over input components) that faithfully replicates the network’s predictions for every possible input masking. Under three common conditions, the conciseness requirement is naturally satisfied by the construction of the AND-OR model, meaning that only a small subset of salient interactions contributes most to the logical model’s total output.

The authors assume that if the interactions in the AND-OR model generalize well to the test set (i.e. appear in the AND-OR model of base model trained on the test samples), the neural network itself also generalizes well to the test set. Building on this claim, one can infer the model’s generalization capability (also during training) from that of an AND-OR model’s interactions. To compute a sufficient rigorous lower bound on the generalization capacity of the logical model, only salient interactions need to be considered, making the generalization test (for both the AND-OR model and the neural network) computationally tractable.

The paper provides a comprehensive evaluation across multiple models, dividing the training process into three stages, where generalization first improves and then declines. The decline corresponds to overfitting - when learned interactions become specific to the training set and fail to generalize to the test set - resulting in decreasing training loss but increasing testing loss. Finally, the authors propose a training method designed to prevent the model from learning non-generalizable interactions, showing that while training loss increases, testing loss remains stable or even improves.

**Strengths:**

1. Novelty: The paper presents a new and non-trivial approach to analyzing the training process and offers an interesting perspective on representing and handling overfitting.

2. Comprehensive Evaluation: The experiments are conducted on six medium+ models trained on standard benchmarks across both NLP and vision domains.

3. Soundness: The experimental results (Fig. 3, Fig. 4) clearly support the main claims of the paper.

4. Scalability / Extendability: The method appears to scale well and can be applied in multiple settings.

**Weaknesses:**

1. Readability: While perhaps inevitable given the topic’s complexity, the paper is difficult to follow. Some provided illustrations (e.g. Fig. 1, Fig. 2) and examples (Fig. 7) are overly complex. Furthermore, the absence of an Evaluation section results in an excessively long and dense Methodology section. The paper also lacks details on how the AND-OR logical model is constructed, which appears to be a very relevant background for this work.

2. Limitations: The proposed method cannot quantify the model’s generalization power with respect to unseen input samples - only for those within the test set.

3. Runtime: The suggested training process to avoid overfitting appears to be time-consuming. According to Eq. (11) and Eq. (12), an AND-OR logical model must be generated for each input sample in the training set, and the generalizability of each salient interaction must be tested against the test data. What is the computational complexity and runtime of this process?

**Questions:**

1. Splitting the paper into separate Methodology and Evaluation sections is strongly recommended.
2. Why do the authors divide the training process into three phases? A more natural split would be into two parts - Phase 1 for increasing generalization power and Phase 2 for decreasing generalization power.
3. Are the three assumptions of the conciseness property really common (all at the same time) in practice?
4. The assumption that the model’s generalization can be derived from the generalization of an AND-OR model's interactions is unclear to me: the latter holds for a specific sample and the power set of its masking options, but not necessarily for all inputs.
5. Adding a simple running example would improve readability. I assume it may be too complicated with current space limitations, so it is only a nice-to-have addition.

Typos:
- Line 431: (dose not exert -> does).
- Line 655: missing $\sigma_S$ in the equation of $o^{or}_S$ (after the minus).
- Line 764: red color for testing loss, blue color for training loss.

---

> ### Author Response · Authors · 2025-11-21
> **Rebuttal by Authors part1**
>
> We sincerely appreciate your valuable comments and are pleased to provide responses to all your questions. In addition, we have carefully revised the manuscript in accordance with your suggestions.
>
> **Should you have any further questions or require additional clarification, please do not hesitate to let us know. We will be glad to follow up as soon as possible.**
>
> ----
>
>
>
> ## Q1: Readability: While perhaps inevitable given the topic’s complexity, the paper is difficult to follow.
>
> > "Some provided illustrations (e.g. Fig. 1, Fig. 2) and examples (Fig. 7) are overly complex. Furthermore, the absence of an Evaluation section results in an excessively long and dense Methodology section. The paper also lacks details on how the AND-OR logical model is constructed, which appears to be a very relevant background for this work"
>
> **A**: Thank you for your understanding. Indeed, investigating the generalization of neural networks from the level of fundamental mechanisms requires the development of an entirely new mathematical framework, which inevitably increases the complexity of the text. We have strived to simplify the explanations and include intuitive diagrams to make the content more accessible. However, the introduction of new theoretical concepts often inherently raises the barrier to comprehension. We will carefully consider your feedback and further refine the language. We believe that an explanation grounded in underlying mechanisms will fundamentally reshape the understanding of generalization in deep neural networks (DNNs), and potential future applications will help more researchers become familiar with this interaction-based perspective.
>
> Additionally, we have restructured the manuscript to enhance clarity. More detailed steps regarding the construction of the logical model is provided in Appendix A.
>
> We sincerely appreciate your encouragement.
>
>
> ----
>
> ## Q2: The proposed method cannot quantify the model’s generalization power with respect to unseen input samples - only for those within the test set.
>
> > "Limitations: The proposed method cannot quantify the model’s generalization power with respect to unseen input samples - only for those within the test set"
>
> **A**: Thank you for your feedback. Our metric for evaluating the generalization power of interactions can, in theory, be extended to accommodate any number of unseen samples. Specifically, it is only necessary to identify the baseline model corresponding to these unseen samples. Thereafter, we can quantify the model's generalization performance with respect to the unseen input data.
>
>
> ----
>
> ## Q3: Runtime: The suggested training process to avoid overfitting appears to be time-consuming. What is the computational complexity and runtime of this process?
>
> > "Runtime: The suggested training process to avoid overfitting appears to be time-consuming. According to Eq. (11) and Eq. (12), an AND-OR logical model must be generated for each input sample in the training set, and the generalizability of each salient interaction must be tested against the test data. What is the computational complexity and runtime of this process?"
>
> **A**: This is an good question, which we will address from the following three perspectives.
>
>
> 1. For each sample, approximately one hundred interactions can be extracted. Thus, only 100 samples are required to plot a histogram of complexity and generalization covering nearly 10,000 interactions. A set of 10,000 interactions is already sufficiently large to represent the quality of the internal representations of the DNN. Therefore, compared to traditional evaluations based on output results, the explanation grounded in interaction mechanisms requires fewer test samples.
>
> 2. The algorithm for computing interactions exhibits exponential computational complexity with respect to the number of input variables. As a result, testing 100 samples on a single RTX 4090 GPU takes approximately 50 minutes.
>
> 3. To mitigate computational complexity, we limit the number of input variables to n ≤ 12. Fortunately, through a series of technical measures, we can ensure that even with n ≤ 12, the majority of information within the neural network can be retained:
>     - For image data, we can partition the image into patches and treat each patch as a single input variable. And we have presented some examples in Appendix G.
>     - For NLP data, we can group multiple tokens into a phrase and annotate the entire group as a single input variable. Similarly, entire sentences or even paragraphs can also be treated as a single variable when dealing with lengthy input prompts. And we have presented some examples in Appendix G.

---

> ### Author Response · Authors · 2025-11-21
> **Rebuttal by Authors part2**
>
> ## Q4: Splitting the paper into separate Methodology and Evaluation sections is strongly recommended.
>
> >"Splitting the paper into separate Methodology and Evaluation sections is strongly recommended."
>
> **A**: Thank you for your suggestion. And we have split the paper into separate Methodology and Evaluation sections.
>
> ---
>
> ## Q5: Why do the authors divide the training process into three phases?
>
> >"Why do the authors divide the training process into three phases? A more natural split would be into two parts - Phase 1 for increasing generalization power and Phase 2 for decreasing generalization power."
>
> **A**: A good question. Indeed, the three-phase dynamics of generalization is delineated based on the pattern of change in the overall performance of the DNN. This overall performance should account for both the number of interactions and the average generalization power of interactions.
>
> Accordingly, the three phases are characterized as follows:
>
> (1) In the first phase, the number of interactions decreases, while their average generalization power increases, leading to an improvement in the overall model performance.
>
> (2) In the second phase, the number of interactions increases and their average generalization power decreases, yet the training-test loss gap begins to widen.
>
> (3) In the third phase, the number of interactions increases and their average generalization power decreases, and the overall model performance decline.
>
> During the second phase, the DNN learns both generalizable interactions (primarily low-order) and non-generalizable interactions (mainly mid- to high-order) simultaneously. In contrast, in the third phase, the learning of high-order interactions becomes dominant—and such high-order interactions are often non-generalizable.
>
> ----
>
> ## Q6：Are the three assumptions of the conciseness property really common (all at the same time) in practice?
>
> > "Are the three assumptions of the conciseness property really common (all at the same time) in practice?"
>
> **A**: Yes, the three assumptions of the conciseness property are common in practice. Furthermore, this conciseness has been observed in prior works, including [cite 1], [cite 2], [cite 3], and [cite 4].
>
> ----
>
> [cite 1] Where We Have Arrived in Proving the Emergence of Sparse Interaction Primitives in DNNs. ICLR, 2024.
>
> [cite 2] Layerwise Change of Knowledge in Neural Networks. ICML, 2024.
>
> [cite 3] Explaining Generalization Power of a DNN using Interactive Concepts. AAAI, 2024.
>
> [cite 4] Defining and quantifying the emergence of sparse concepts in dnns. CVPR, 2023.

---

> ### Author Response · Authors · 2025-11-21
> **Rebuttal by Authors part3**
>
> ## Q7：The assumption that the model’s generalization can be derived from the generalization of an AND-OR model's interactions is unclear to me: the latter holds for a specific sample and the power set of its masking options, but not necessarily for all inputs
>
> >"The assumption that the model’s generalization can be derived from the generalization of an AND-OR model's interactions is unclear to me: the latter holds for a specific sample and the power set of its masking options, but not necessarily for all inputs. "
>
> **A**: A: That is a good question. For a given input sample, its output classification score can be decomposed into the sum of different interactions, which includes both generalizable and non-generalizable interactions. While different input samples activate distinct sets of interactions, these distinct sets of interactions can still be categorized into generalizable and non-generalizable types. Our theoretical framework does not assume that any two input samples must share similar interactions; instead, it emphasizes whether the triggered interactions can generalize to certain unseen test samples. Specifically, the histograms of generalizable and non-generalizable interactions presented in Figure 3 represent averaged statistical outcomes derived from 100 input samples and roughly 10,000 different interactions per DNN. Thus, we can conclude that although different input samples typically produce different interactions, the resulting classification scores can still be interpreted through the generalization capability of these interactions.
>
> ----
>
> ## Q8：Adding a simple running example would improve readability.
>
> >"Adding a simple running example would improve readability. I assume it may be too complicated with current space limitations, so it is only a nice-to-have addition"
>
> **A**: Thank you for your suggestion. In the Supplementary Material, we have included a video demo that we hope offers a clearer, more intuitive illustration of symbolic interaction. Please feel free to watch it at your convenience.
>
> ----
>
> ## Q9: About typo.
>
> > "	Line 431: (dose not exert -> does).
> 	Line 655: missing \sigma_S in the equation of o_S^{or} (after the minus).
> 	Line 764: red color for testing loss, blue color for training loss.
> "
>
> **A**: A: Thank you for your suggestion. We have followed your suggestions to polish our manuscript. It is worth mentioning that at line 431, we employ the phrase "does not exert" to indicate that if an interaction fails to meet this condition, it should be regarded as non-generalizable.
>
> ----
>
> **Please feel free to reach out with any further questions. We look forward to your feedbacks and will reply promptly.**

---

### Official Review · Reviewer_kUqg · 2025-10-31

**Soundness:** 3
**Presentation:** 3
**Contribution:** 3
**Rating:** 6
**Confidence:** 3

**Summary:**

This paper investigates the generalization dynamics of deep neural networks (DNNs) through the lens of symbolic interactions. Building on prior work showing that DNN inference can be decomposed into AND-OR interaction patterns, the authors propose a method to quantify the generalization power of individual interactions by comparing them against baseline DNNs trained on test data. They discover a three-phase training dynamic: (1) early phase removes non-generalizable interactions and learns simple patterns, (2) middle phase learns increasingly complex interactions with decreasing generalization, and (3) late phase predominantly learns non-generalizable interactions leading to overfitting. Experiments on vision and NLP datasets validate these findings.

**Strengths:**

**Comprehensive theoretical foundation**
The paper builds systematically on the AND-OR interaction framework from prior work, providing clear mathematical formulations and proofs.

**Well-articulated three-phase dynamics**
The identification of distinct training phases characterized by interaction generalization power provides new insights into DNN training dynamics and offers a principled explanation for the train-test gap.

**Extensive empirical validation**
Experiments span multiple architectures and datasets, demonstrating consistency of findings across domains.

**Weaknesses:**

**Unclear causality**
While the paper shows correlation between non-generalizable interactions and overfitting, the causal direction is unclear. Are non-generalizable interactions the cause of overfitting, or merely a symptom of models memorizing training data?

**Phase transitions not rigorously defined.** The three phases are identified qualitatively from figures, but there are no precise criteria or automatic methods to detect phase boundaries. This makes the framework difficult to apply systematically.

**Minor presentation issues.**
There’s a typo (“non-gneralizable”) in Fig. 4.

**Questions:**

1. How can the method scale beyond 10 input variables? Can this framework be extended to realistic input dimensions?

2. Why is training on test data a valid approach for identifying generalizable interactions? Could alternative methods (e.g., cross-validation set) achieve similar results without test data access?

3. How do findings extend to attention-based models, where interactions might be explicitly modeled through attention mechanisms?

---

> ### Author Response · Authors · 2025-11-21
> **Rebuttal by Authors part1**
>
> Thank you for your thorough review and valuable feedback. We will address all your questions comprehensively.
>
> **Should you have any follow-up questions or need further clarification, please don't hesitate to contact us**
>
> ----
>
> ## Q1: Unclear causality. While the paper shows correlation between non-generalizable interactions and overfitting, the causal direction is unclear. Are non-generalizable interactions the cause of overfitting, or merely a symptom of models memorizing training data?
>
> >"Unclear causality. ... overfitting, the causal direction is unclear. Are non-generalizable interactions the cause of overfitting, or merely a symptom of models memorizing training data?"
>
> **A**: A good question. Let us answer your question from the following four perspectives.
>
> First, how to define generalizability. We define a generalizable interaction as follows:
>
> We quantify the generalizability of individual interactions encoded by the target DNN to a baseline DNN, which is denoted by $v^{\\text{base}}$ and trained on the testing samples (please refer to Appendix E.1 for details). To elucidate this approach, let us consider an extracted salient AND interaction $S$, s.t. $|I^{\\text{and}}\_S| > \\tau$. To this end, the interaction $S$ is deemed to represent an inference pattern that can generalize to the testing set, if the baseline DNN $v^{\\text{base}}$ also extracts the same AND interaction $S$ with a salient effect (i.e., $|I^{\\text{and}}\_{S,v^{\\text{base}}}| > \\tau$) from the input, and the AND interaction $S$ exerts a consistent directional influence on the classification of $\\mathbf{x}$ (i.e., $I^{\\text{and}}\_{S,v^{\\text{base}}} \\cdot I^{\\text{and}}\_S > 0$). The same principle also applies to OR interactions.  The generalizability of an AND/OR interaction is assessed by the following metric:
>
> $$
> \\begin{aligned}
>     \\mathcal{G}^{\\text{type}}\_{S,v^{\\text{base}}} &= \\mathbf{1}\\big(\\lvert I^{\\text{type}}\_{S,v^{\\text{base}}}\\rvert > \\tau\\big)\\cdot \\mathbf{1}\\big(I^{\\text{type}}\_S \\cdot I^{\\text{type}}\_{S,v^{\\text{base}}} > 0\\big) \\in \\{0,1\\},
> \\end{aligned}
> $$
>
> where $\\mathbf{1}(\\cdot)$ is a binary indicator function that outputs $1$ only when the condition is satisfied, $\\text{type} \\in \\{\\mathrm{and}, \\mathrm{or}\\}$.
>
> To this end, when metric $\\mathcal{G}^{\\text{and}}\_{S,v^{\\text{base}}}=1$ ，interaction $S$ is considered to be a generalizable interaction; otherwise $S$ is considered to be a non-generalizable interaction.
>
> This definition does not distinguish whether the interaction is merely memorization of the training data, or a more essential generalizable “knowledge” beyond memorization. Notably, there is no clear boundary between concrete memories of input samples and metaphysically essential knowledge in practice, because natural language texts often directly state certain knowledge, and in such cases, the memorized text is generalizable knowledge. Therefore, we merely propose the metric $\mathcal{G}$ to quantify the generalizability of each interaction.
>
> Second, all interactions, including generalizable ones and non-generalizable ones, are learned to reduce the training loss, because they are all automatically encoded by the DNN during the training process. Therefore, if the learned interaction cannot help the inference on testing samples, then this interaction can be considered to cause overfitting --- only helping minimize the training loss but not helping the classification of testing samples. From this perspective, this is not agnostic to whether the interaction represents “a symptom of models memorizing training data.”
>
> Third, based on interactions, we can find lots of examples of how simple memorization of training data causes overfitting. For example, when an LLM simply memorizes the input tokens and the statistical co-occurrence relationships of input tokens in prompts, without gaining an in-depth understanding of the mechanisms behind reasoning, such memorization can be regarded as overfitting to the training data.
>
> Fourth, defining causality for internal interaction patterns remains an open problem. Although the interactions discussed in this paper do affect the DNN’s outputs—and can, from an input–output function mapping perspective, be viewed as partial causal factors—they do not satisfy the criteria of rigorous causality in conventional causal analysis. This discrepancy arises because prior research has primarily addressed causality in the physical world, rather than within the input–output mappings defined by DNNs. And in physical world, a cause must directly lead to its effect, temporally precede it, and generally adhere to established causal laws. These requirements, however, lie beyond the scope of this study. Moreover, it is generally infeasible to formulate a mathematical definition that encompasses all factors extending beyond the DNN’s input-to-output functional mapping.

---

> ### Author Response · Authors · 2025-11-21
> **Rebuttal by Authors part2**
>
> ## Q2: Phase transitions not rigorously defined. The three phases are identified qualitatively from figures, but there are no precise criteria or automatic methods to detect phase boundaries. This makes the framework difficult to apply systematically
>
> >"Phase transitions not rigorously defined. The three phases are identified qualitatively from figures, but there are no precise criteria or automatic methods to detect phase boundaries. This makes the framework difficult to apply systematically"
>
> **A**:  We appreciate your suggestion. Adopting a more rigorous quantitative approach indeed helps to better delineate the boundaries between the three phases. Following your advice, we have formulated more precise, quantitative definitions for these phases below.
>
> We characterize the learning dynamics using several key metrics. Let $\delta\_g(t)$ and $\delta\_n(t)$ be the incremental strengths of new **generalizable** and **non-generalizable** interactions learned at time $t$. We also track the average interaction generalization power, $\\overline{\\mathcal{H}}(t)$, and the number of learned interactions, $\\overline{\\mathcal{N}}(t)$.
>
> We can now characterize the three phases:
>
> * **Phase 1 (Interval: $[t\_0, t\_1]$)**
>     The model primarily learns generalizable patterns. $\\overline{\\mathcal{H}}(t)$ increases while $\\overline{\\mathcal{N}}(t)$ decreases. This phase concludes when $\\overline{\\mathcal{H}}(t)$ reaches its peak.
>     $$
>     t\_1 = \\underset{t}{\\operatorname{argmax}} \\ \\overline{\\mathcal{H}}(t)
>     $$
>
> * **Phase 2 (Interval: $(t\_1, t\_2]$)**
>     The model begins to learn non-generalizable interactions along with generalizable interactions. $\\overline{\\mathcal{H}}(t)$ decreases and $\\overline{\\mathcal{N}}(t)$ increases. Learning of generalizable patterns still dominates, satisfying the condition $\\delta\_g(t) \\ge r \\cdot \\delta\_n(t)$. This phase ends at time $t\_2$, the first point where the strengths balance: $\\delta\_g(t) = r \\cdot \\delta\_n(t)$, given the monotonically increasing of $\\frac{\\Delta\_n(t)}{\\Delta\_g(t)}$.
>
> * **Phase 3 (Interval: $(t\_2, t\_3]$)**
>     The acquisition of new interactions is now dominated by non-generalizable ones, with their strengths satisfying $\\delta\_g(t) < r \\cdot \\delta\_n(t)$.  $\\overline{\\mathcal{N}}(t)$ continues to increase and $\\overline{\\mathcal{H}}(t)$ continues to decrease.
>
>
> Based on the above definition, we observed that the new phase demarcation remained consistent with the original partitioning described in our manuscript, which further validated the generalization dynamics identified in our work. We have updated this definition to Appendix I.
>
> ----
>
> ## Q3: Minor presentation issues
>
> > "Minor presentation issues. There’s a typo (“non-gneralizable”) in Fig. 4."
>
> **A**: Thank you for your comments. We have followed your suggestions to carefully polish the language.
>
> ---
>
> ## Q4: How can the method scale beyond 10 input variables? Can this framework be extended to realistic input dimensions?
>
> >"How can the method scale beyond 10 input variables? Can this framework be extended to realistic input dimensions?"
>
> A: A good question. In principle, our method is a generic algorithm that can be applied to DNNs with different numbers of input variables. However, due to the high computational cost, we typically restrict the number of input variables in practice. This may somewhat influence the discovery of interactions and cross-task comparability. However, such limitations can be mitigated through several engineering strategies:
>
> Strategy 1: For image data, we can partition the image into patches and treat each patch as a single input variable. And we have presented some examples in Appendix G.
>
> Strategy 2: For NLP data, we can group multiple tokens into a phrase and annotate the entire group as a single input variable. Similarly, an entire sentence or even a paragraph can also be treated as a single variable when dealing with lengthy input prompts. And we have presented some examples in Appendix G.
>
> These strategies make our method applicable to most applications without causing a prohibitive computational cost.

---

> ### Author Response · Authors · 2025-11-21
> **Rebuttal by Authors part3**
>
> ## Q5: Why is training on test data a valid approach for identifying generalizable interactions? Could alternative methods (e.g., cross-validation set) achieve similar results without test data access?
>
> >"Why is training on test data a valid approach for identifying generalizable interactions? Could alternative methods (e.g., cross-validation set) achieve similar results without test data access?"
>
> **A**: A good question. We have followed your suggestions to conduct the cross-validation to examine the faithfulness of the extracted generalizable interactions.
>
> Specifically, we uniformly partitioned all data into three sets (A, B, and C) to perform cross-validation. We trained the target DNN on the dataset A, and trained two baseline DNNs on the dataset B and the dataset C, respectively. Here, we regarded the baseline model trained on dataset B as representing the interactions on the validation set and regarded the baseline model trained on dataset C as representing the interactions on the test set. Then, given each input sample, we followed Equation (7) in Section 2.2 to measure the distribution of interactions that could generalize to the validation set (the set B), i.e., \\(\\mathbb{I}\_\{\\text{gen}\_\\text{B}\}^{(m),+}\\) and \\(\\mathbb{I}\_\{\\text{gen}\_\\text{B}\}^{(m),-}\\). We also measured the distribution of interactions that could generalize to the test set (the set C), i.e., \\(\\mathbb{I}\_\{\\text{gen}\_\\text{C}\}^{(m),+}\\) and \\(\\mathbb{I}\_\{\\text{gen}\_\\text{C}\}^{(m),-}\\).[^1]
>
> The validation was conducted to test whether the interactions, which could generalize to the validation set, could also generalize to the test set. If feasible, the faithfulness of our method was validated. Specifically, we computed the Jaccard similarity between the two distributions as follows:
>
> \\[
> J = \\frac{ \\displaystyle \\sum\_\{m=1\}^{n} \\Big[ \\min\\!\\big(\\,\\left|\\mathbb{I}\_\{\\text{gen}\_\\text{B}\}^{(m),+}\\right|,\\,\\left|\\mathbb{I}\_\{\\text{gen}\_\\text{C}\}^{(m),+}\\right|\\big) + \\min\\!\\big(\\,\\left|\\mathbb{I}\_\{\\text{gen}\_\\text{B}\}^{(m),-}\\right|,\\,\\left|\\mathbb{I}\_\{\\text{gen}\_\\text{C}\}^{(m),-}\\right|\\big) \\Big] }{ \\displaystyle \\sum\_\{m=1\}^{n} \\Big[ \\max\\!\\big(\\,\\left|\\mathbb{I}\_\{\\text{gen}\_\\text{B}\}^{(m),+}\\right|,\\,\\left|\\mathbb{I}\_\{\\text{gen}\_\\text{C}\}^{(m),+}\\right|\\big) + \\max\\!\\big(\\,\\left|\\mathbb{I}\_\{\\text{gen}\_\\text{B}\}^{(m),-}\\right|,\\,\\left|\\mathbb{I}\_\{\\text{gen}\_\\text{C}\}^{(m),-}\\right|\\big) \\Big] }
> \\]
>
> The experimental results showed that the DNNs exhibited high Jaccard similarity of generalizable interaction distribution between the test set and the validation set. This supported the faithfulness of our method of identifying generalizable interactions.
>
> We have added the detailed experimental settings and results in the newly added Appendix N.
>
>
> [^1]: Specifically, extending the definition in Equation (7), we utilized \\(\\mathbb{I}\_\{\\text{gen}\_\\text{B}\}^{(m),+}\\) to denote the aggregate effect strength of all generalizable salient interactions with **positive** effect extracted from the baseline DNN trained on dataset B, and \\(\\mathbb{I}\_\{\\text{gen}\_\\text{B}\}^{(m),-}\\) to denote the aggregate effect strength of all generalizable salient interactions with **negative** effect extracted from the baseline DNN trained on dataset B. Similarly, \\(\\mathbb{I}\_\{\\text{gen}\_\\text{C}\}^{(m),+}\\) and \\(\\mathbb{I}\_\{\\text{gen}\_\\text{C}\}^{(m),-}\\) denote the aggregate effect strengths (positive and negative, respectively) of generalizable salient interactions extracted from the baseline DNN trained on dataset C.

---

> ### Author Response · Authors · 2025-11-21
> **Rebuttal by Authors part4**
>
> ## Q6: How do findings extend to attention-based models, where interactions might be explicitly modeled through attention mechanisms?
>
> >"How do findings extend to attention-based models, where interactions might be explicitly modeled through attention mechanisms?"
>
> A: Thank you for your comments. In fact, our experiments in Figure 3 have already shown results on attention-based DNNs. The BERT-medium model and the BERT-large model in Figure 3 are two typical attention-based DNNs. The experimental results have shown that the distribution of interactions follows a three-phase dynamics during training process of the DNN.
>
> More importantly, we have also conducted a **new experiment** on several open source LLMs based on the transformer (attention) architecture, including Qwen2.5-7B, and DeepSeek-R1-Distill-Llama-8B, which were well-known large language models and have been widely applied.
>
> Notably, we acknowledged the significant challenge of retraining a foundational model from scratch to serve as a baseline. Therefore, considering that different LLMs released by various organizations/companies were typically trained on different and massive datasets, we provisionally utilized one LLM as the baseline DNN to evaluate another.
>
> Specifically, we selected Qwen2.5-7B model as the baseline model and DeepSeek-R1-Distill-Llama-8B model as the target model. We used data from the SQuAD dataset and followed Equation (7) from Section 2.2 in the manuscript to measure the distribution of interactions that could generalize to the baseline DNN.
>
> The **new experimental results** suggested that the two LLMs tended to encode low-order (simple) generalizable interactions. This indicated that these well-trained LLMs might focus on learning simple patterns during training. And such simple interactions could potentially be more transferable between models.
>
> Please see the newly added Appendix O for results.
>
> -----
>
> **For any further questions or clarifications, please feel free to contact us. We will respond as soon as possible.**

---

> ### Author Response · Authors · 2025-11-25
> **Follow-Up on Rebuttal**
>
> Dear Reviewer kUqg,
>
> We hope this message finds you well. For any further questions or clarifications, please feel free to contact us. We will respond as soon as possible.
>
> Best regards,
>
> The authors

---

> > ### Comment · Reviewer_kUqg · 2025-11-27
> >
> > I thank the authors for their detailed response. The comments have addressed my concerns and I will maintain my positive score.

---

### Official Review · Reviewer_NKef · 2025-11-03

**Soundness:** 3
**Presentation:** 3
**Contribution:** 3
**Rating:** 6
**Confidence:** 1

**Summary:**

This submission proposes a framework to define, quantify, and track the generalization power of primitive inference patterns—formalized as AND-OR “interactions” among input variables—encoded by deep neural networks (DNNs) over the course of training.  The paper shows that removing non-generalizable interactions from the network’s output narrows the train–test loss gap, suggesting these interactions are a proximate cause of overfitting.

**Strengths:**

1. The paper is clearly written with helpful figures and a step-by-step methodology.
2. The framework is anchored in prior formal results ensuring exact matching on masked inputs and sparsity of salient interactions under common conditions.
3. The three-phase dynamic is intuitively appealing and coheres with known overfitting behavior, offering a clear diagnostic narrative for training curves.

**Weaknesses:**

1. The binary label G may be too coarse for nuanced interactions.
2. Claims of causality are suggestive but not definitive: removing non-generalizable terms is consistent with overfitting, yet residual confounds may remain.
3. The baseline DNN is trained on the test set, which could bias labels and inflate H if the baseline overfits
4. The masking protocol and the restriction to exactly 10 variables per input may affect interaction discovery and comparability across tasks

**Questions:**

For LLMs, can you extend analyses from masked classification settings to generation (e.g., next-token distributions) with interaction tracking over context length? What changes in the three-phase dynamics do you expect?

Is there evidence of rare but truly generalizable high-order interactions (e.g., compositional patterns)? How would the current proxy distinguish them from non-generalizable ones?

---

> ### Author Response · Authors · 2025-11-21
> **Rebuttal by Authors part1**
>
> Thank you for your review and feedback. We are committed to addressing all of your questions.
>
> **Should any points require further clarification, please do not hesitate to inform us promptly.**
>
> ---
>
> ## Q1: The binary label G may be too coarse for nuanced interactions
> > "... G may be too coarse ..."
>
> Thank you for your comments. In fact, we have followed your suggestions to redefine the metric for the interaction’s generalizability within the range \([0,1]\). Specifically, for each salient **AND** interaction \(S\) (or each salient **OR** interaction \(S\)), we have redefined its generalization power as the similarity between the interaction effect in the target model \(v\) and that in the baseline model $v^{\\mathrm{base}}$, as follows:
>
> $$
> \\mathcal{G}'^{\\mathrm{and}}\_{S,v_{\\mathrm{base}}}
> = \\mathbf{1}\\!\\big(I_S^{\\mathrm{and}}\\cdot I_{S,v_{\\mathrm{base}}}^{\\mathrm{and}}>0\\big)
> \\;\\cdot\\;
> \\frac{\\min\\!\\big(|I_S^{\\mathrm{and}}|,\\;|I_{S,v_{\\mathrm{base}}}^{\\mathrm{and}}|\\big)}
> {\\max\\!\\big(|I_S^{\\mathrm{and}}|,\\;|I_{S,v_{\\mathrm{base}}}^{\\mathrm{and}}|\\big)} ,
> $$
>
> $$
> \\mathcal{G}'^{\\mathrm{or}}\_{S,v_{\\mathrm{base}}}
> = \\mathbf{1}\\!\\big(I_S^{\\mathrm{or}}\\cdot I_{S,v_{\\mathrm{base}}}^{\\mathrm{or}}>0\\big)
> \\;\\cdot\\;
> \\frac{\\min\\!\\big(|I_S^{\\mathrm{or}}|,\\;|I_{S,v_{\\mathrm{base}}}^{\\mathrm{or}}|\\big)}
> {\\max\\!\\big(|I_S^{\\mathrm{or}}|,\\;|I_{S,v_{\\mathrm{base}}}^{\\mathrm{or}}|\\big)} .
> $$
>
> Here, \\(\\mathbf{1}(\\cdot)\\) is an indicator function which returns \\(1\\) if the condition holds and \\(0\\) otherwise.
> The ratio term $\\frac{\\min\\big(|I_S|,\\;|I_{S,v^{\\mathrm{base}}}|\\big)}{\\max\\big(|I_S|,\\;|I_{S,v^{\\mathrm{base}}}|\\big)} $ measures the similarity of the interaction effects encoded by the two models and therefore reflects the generalizability of this interaction. Only when the same interaction \(S\) (for AND or OR) is extracted by both DNNs and exhibits exactly the same effect do we consider it to be \(100\%\) generalizable.
>
> Notably, because outputs of different DNNs often have different ranges of perturbation, we further normalize the interaction effects. Concretely, we normalize
> the AND/OR interaction effects as
> \\(I^{\\mathrm{and}}\_{S} \\leftarrow
> \\frac{I^{\\mathrm{and}}\_{S}}{\\mathbb{E}\_x\\big[\\,|v(N)-v(\\varnothing)|\\,\\big]}\\), and \\(I^{\\mathrm{or}}\_{S} \\leftarrow \\frac{I^{\\mathrm{or}}\_{S}}{\\mathbb{E}\_x\\big[\\,|v(N)-v(\\varnothing)|\\,\\big]}\\)
> .
>
> Using these redefined metrics $\\mathcal{G}'^{\\mathrm{and}}\_{S,v_{\\mathrm{base}}}$ and $\\mathcal{G}'^{\\mathrm{or}}\_{S,v_{\\mathrm{base}}}$, we conducted **extensive experiments** to analyze the distribution of generalizable interactions across different interaction orders. The new results align closely with results based on the old metric. Because the new metric is more stringent, the strength of generalizable interactions is reduced; however, the distribution of generalizable interactions over different orders based on the new metric remains similar to that based on the old metric. Please see Appendix M for the new results.
>
> ----
>
> ## Q2: Claims of causality are suggestive but not definitive: removing non-generalizable terms is consistent with overfitting, yet residual confounds may remain
>
> > "... not definitive: removing non-generalizable ...confounds may remain"
>
> **A**: Thank you for this comment. We have followed your suggestion to revise the original claim ”these results confirm that the removed non-generalizable interactions were the fundamental cause for overfitting” to the new one “these results suggest that the removed non-generalizable interactions were highly correlated with the overfitting of the DNN.”
> In fact, the strong correlation between the encoding of non-generalizable interactions and the over-fitting of the DNN can be obtained from the following two perspectives:
> 1. According to experimental results in Figure 4, it has been observed that the removal of non-generalizable interactions has effectively reduced the training-testing loss gap. Please see Section 3.2 for experimental details.
> 2. Experimental results in Figure 3 show that the three-phase dynamics of generalization power of interactions are strictly aligned with the overfitting progression of the DNN. The start of Phase 2 aligns with the onset of the widening of the DNN’s training-testing loss gap. It indicates that the gradual learning of interactions with increasing orders may be the internal mechanism for the widening of the training-testing loss gap. This also well fits people’s intuition that simple interactions (i.e., low-order ones) are more likely to represent meaningful/generalizable features. Please see Section 3.1 for details.
>
> Nevertheless, we still cannot completely rule out other factors that may also cause neural network overfitting. Therefore, we have modified our conclusions in the manuscript to more carefully reflect the suggestive nature of this correlation.
>
> ----

---

> ### Author Response · Authors · 2025-11-21
> **Rebuttal by Authors part2**
>
> ## Q3: The baseline DNN is trained on the test set, which could bias labels and inflate H if the baseline overfits.
>
> >"The baseline DNN is trained on the test set which could bias labels and inflate H if the baseline overfits."
>
> **A**: A good question. We agree that training the baseline model cannot completely avoid potential model biases. To this end, we have conducted a new experiment to explicitly evaluate faithfulness of the quantified generalization power. Please see Appendix N for detailed experimental settings and results.
> Specifically, we uniformly partitioned all data into three sets (A, B, and C) to perform cross-validation. We trained the target DNN on the dataset A, and trained two baseline DNNs on the dataset B and the dataset C, respectively. Here, we regarded the baseline model trained on dataset B as representing the interactions on the validation set and regarded the baseline model trained on dataset C as representing the interactions on the test set. Then, given each input sample, we followed Equation (7) in Section 2.2 to measure the distribution of interactions that could generalize to the validation set (the set B), i.e., \\(\\mathbb{I}\_\{\\text{gen}\_\\text{B}\}^{(m),+}\\) and \\(\\mathbb{I}\_\{\\text{gen}\_\\text{B}\}^{(m),-}\\). We also measured the distribution of interactions that could generalize to the test set (the set C), i.e., \\(\\mathbb{I}\_\{\\text{gen}\_\\text{C}\}^{(m),+}\\) and \\(\\mathbb{I}\_\{\\text{gen}\_\\text{C}\}^{(m),-}\\).[^1]
>
> The validation was conducted to test whether the interactions, which could generalize to the validation set, could also generalize to the test set. If feasible, the faithfulness of our method was validated. Specifically, we computed the Jaccard similarity between the two distributions as follows:
>
> \\[
> J = \\frac{ \\displaystyle \\sum\_\{m=1\}^{n} \\Big[ \\min\\!\\big(\\,\\left|\\mathbb{I}\_\{\\text{gen}\_\\text{B}\}^{(m),+}\\right|,\\,\\left|\\mathbb{I}\_\{\\text{gen}\_\\text{C}\}^{(m),+}\\right|\\big) + \\min\\!\\big(\\,\\left|\\mathbb{I}\_\{\\text{gen}\_\\text{B}\}^{(m),-}\\right|,\\,\\left|\\mathbb{I}\_\{\\text{gen}\_\\text{C}\}^{(m),-}\\right|\\big) \\Big] }{ \\displaystyle \\sum\_\{m=1\}^{n} \\Big[ \\max\\!\\big(\\,\\left|\\mathbb{I}\_\{\\text{gen}\_\\text{B}\}^{(m),+}\\right|,\\,\\left|\\mathbb{I}\_\{\\text{gen}\_\\text{C}\}^{(m),+}\\right|\\big) + \\max\\!\\big(\\,\\left|\\mathbb{I}\_\{\\text{gen}\_\\text{B}\}^{(m),-}\\right|,\\,\\left|\\mathbb{I}\_\{\\text{gen}\_\\text{C}\}^{(m),-}\\right|\\big) \\Big] }
> \\]
>
> The experimental results showed that the DNNs exhibited high Jaccard similarity of generalizable interaction distribution between the test set and the validation set. This supported the faithfulness of our method of identifying generalizable interactions.
>
> We have added the detailed experimental settings and results in the newly added Appendix N.
>
>
> [^1]: Specifically, extending the definition in Equation (7), we utilized \\(\\mathbb{I}\_\{\\text{gen}\_\\text{B}\}^{(m),+}\\) to denote the aggregate effect strength of all generalizable salient interactions with **positive** effect extracted from the baseline DNN trained on dataset B, and \\(\\mathbb{I}\_\{\\text{gen}\_\\text{B}\}^{(m),-}\\) to denote the aggregate effect strength of all generalizable salient interactions with **negative** effect extracted from the baseline DNN trained on dataset B. Similarly, \\(\\mathbb{I}\_\{\\text{gen}\_\\text{C}\}^{(m),+}\\) and \\(\\mathbb{I}\_\{\\text{gen}\_\\text{C}\}^{(m),-}\\) denote the aggregate effect strengths (positive and negative, respectively) of generalizable salient interactions extracted from the baseline DNN trained on dataset C.
>
>
> ---
>
> ## Q4: The masking protocol and the restriction to exactly 10 variables per input may affect interaction discovery and comparability across tasks.
>
> >"he masking protocol and the restriction to exactly 10 variables per input may affect interaction discovery and comparability across tasks."
>
> **A**: A good question. In principle, our method is a generic algorithm that can be applied to DNNs with different numbers of input variables. However, due to the high computational cost, we typically restrict the number of input variables in practice. This may somewhat influence the discovery of interactions and cross-task comparability.
> However, such limitations can be mitigated through several engineering strategies:
>
> Strategy 1: For image data, we can partition the image into patches and treat each patch as a single input variable. And we have presented some examples in Appendix G.
>
> Strategy 2: For NLP data, we can group multiple tokens into a phrase and annotate the entire group as a single input variable. Similarly, an entire sentence or even a paragraph can also be treated as a single variable when dealing with lengthy input prompts. And we have presented some examples in Appendix G.
>
> These strategies make our method applicable to most applications without causing a prohibitive computational cost.

---

> ### Author Response · Authors · 2025-11-21
> **Rebuttal by Authors part3**
>
> ## Q5: For LLMs, can you extend analyses from masked classification settings to generation (e.g., next-token distributions) with interaction tracking over context length? What changes in the three-phase dynamics do you expect?
>
> >"For LLMs, ...to generation with interaction tracking ... changes in the three-phase ..."
>
> **A**: A good question. Our analysis can be extended to next-token distributions. We have conducted **new experiments** on LLMs to analyze interaction generalization power in the newly added Appendix O.
>
> Because of the limited time for rebuttal, we cannot thoroughly execute the entire process of training a foundation LLM from scratch. Instead, let us temporarily focus on the SFT of a trained LLM.
>
> Due to the limited time of the rebuttal, we used Unilaw-R1 Data [cite 1] to fine-tune Qwen2.5-7B-Instruct model , which was originally trained on 18T tokens of data. Experimental results showed that during the SFT process, low-order interactions increased, while high-order interactions decreased. The proportion of generalizable interactions changed from 45.2% to 52.3%. The current experimental results had shown that the SFT process was still within the first phase. We will continue to finetune the LLM and keep reporting latest distribution of interactions to provide dynamics in the second and third phases in future. The above experiments show that our method remains applicable to analyzing LLM training. Please see newly added Appendix P for details.
>
> [cite 1] Unilaw-R1: A Large Language Model for Legal Reasoning with Reinforcement Learning and Iterative Inference. Cai, Hua et al.
>
> ---
>
> ## Q6: Is there evidence of rare but truly generalizable high-order interactions (e.g., compositional patterns)? How would the current proxy distinguish them from non-generalizable ones?
>
> >"... high-order interactions ... would the current proxy distinguish them from non-generalizable ones?"
>
> **A**: A good question. The answer is yes. To illustrate such generalizable high-order interactions, we conducted an experiment on the Qwen-2.5 7B and deepseek-r1-distill-llama-8b. The input prompt was set as “Major advances were registered in research on core technologies in key fields, and breakthroughs were made in manned spaceflight, Mars exploration, resource”, and we analyzed the target prediction ”exploration”.
>
> In this case, we successfully extracted such high-order generalizable interactions. Both models encode interaction between input variables in **S={[Major advances],[ core technologies],[ key fields,],[ manned spaceflight,],[ Mars exploration,],[ resource]}** with salient effects that both contribution positive values to boost the confidence of generating “exploration”. This is a typical example of the generalizable high-order interaction.
>
> Our current method distinguishes such high-order generalizable interactions as follows. We quantify the generalizability of individual interactions encoded by the target DNN to a baseline DNN, which is denoted by $v^{\\text{base}}$ and trained on the testing samples (please refer to Appendix E.1 for details). To elucidate this approach, let us consider an extracted salient AND interaction $S$, s.t. $|I^{\\text{and}}\_S| > \\tau$. To this end, the interaction $S$ is deemed to represent an inference pattern that can generalize to the testing set, if the baseline DNN $v^{\\text{base}}$ also extracts the same AND interaction $S$ with a salient effect (i.e., $|I^{\\text{and}}\_{S,v^{\\text{base}}}| > \\tau$) from the input, and the AND interaction $S$ exerts a consistent directional influence on the classification of $\\mathbf{x}$ (i.e., $I^{\\text{and}}\_{S,v^{\\text{base}}} \\cdot I^{\\text{and}}\_S > 0$). The same principle also applies to OR interactions. Accordingly, the generalizability of an AND/OR interaction is assessed by the following binary metrics:
>
> $$
> \\begin{aligned}
>     \\mathcal{G}^{\\text{and}}\_{S,v^{\\text{base}}} &= \\mathbf{1}\\big(\\lvert I^{\\text{and}}\_{S,v^{\\text{base}}}\\rvert > \\tau\\big)\\cdot \\mathbf{1}\\big(I^{\\text{and}}\_S \\cdot I^{\\text{and}}\_{S,v^{\\text{base}}} > 0\\big) \\in \\{0,1\\},\\\\[6pt]
>     \\mathcal{G}^{\\text{or}}\_{S,v^{\\text{base}}} &= \\mathbf{1}\\big(\\lvert I^{\\text{or}}\_{S,v^{\\text{base}}}\\rvert > \\tau\\big)\\cdot \\mathbf{1}\\big(I^{\\text{or}}\_S \\cdot I^{\\text{or}}\_{S,v^{\\text{base}}} > 0\\big) \\in \\{0,1\\},
> \\end{aligned}
> $$
>
> where $\\mathbf{1}(\\cdot)$ is a binary indicator function that outputs $1$ only when the condition is satisfied.
>
> To this end, when metric $\\mathcal{G}^{\\text{and}}\_{S,v^{\\text{base}}}=1$ (or $\\mathcal{G}^{\\text{or}}\_{S,v^{\\text{base}}}=1$)，interaction $S$ is considered to be a generalizable interaction; otherwise $S$ is considered to be a non-generalizable interaction.
>
>
> **Should you have any additional concerns or require further clarification regarding specific details, please do not hesitate to contact us. We will respond promptly to your inquiries.**

---

> ### Author Response · Authors · 2025-11-25
> **Follow-Up on Rebuttal**
>
> Dear Reviewer NKef，
>
> We hope this message finds you well. Should you have any additional concerns or require further clarification regarding specific details, please do not hesitate to contact us. We will respond promptly to your inquiries.
>
> Best regards,
>
> The authors

---

### Author Response · Authors · 2025-11-21

Dear reviewers, thank you for your patience. We are preparing detailed, point-by-point responses and carrying out the necessary additional experiments; we will submit our replies and updates as soon as they are completed.

---

### Meta-Review · Area_Chair_Cu9R · 2025-12-17

**Summary:**

- This work makes sweeping claims (about universality and causality) without any significant justification. The exposition of the paper is extremely poor. Definitions presented in the paper are suggestive and elusive. Multiple reviewers have also pointed this out.

- During the rebuttal, the authors try to portray their proposal as computational feasible. It turns out this is very inaccurate. Indeed, a somewhat adhoc procedure is proposed in Appendix A.1 to extract "interactions". The pseudo code (Algorithm 1, which forms a backbone of the proposal) contains multiple snippets which go like so: "for every subset S of $\{1,2,...,n\}$, do something", and therefore has running time at leadt $\Omega(2^n)$, where $n$ is the the number of "input variables". This is a NO NO, and is enough to kill the paper completely. The authors then make some adhoc choices which ensures they only look at $n \le 12$(or 10, actually) input variables. Why not (?), but then it is not clear what's the justification (also pointed out by **Reviewer NKef**, **Reviewer kUqg**, and **Reviewer K1xr**) nor what the paper is trying to do from this point on...


- Even if we put aside the above obvious practical/computational issues, the paper makes no tangible theoretical contribution. Therefore, I find no way to save the paper neither from an empirical nor theoretical front.

For the above and many other reasons, I recommend the paper be rejected. I'd advice the authors integrate the above comments and the longer comments from the reviewers, and completely redo the message of their manuscript, the presentation, and justification of their claims, before resubmitting.

**Reviewer Concerns:**

The main issue not satisfactorily addressed by the rebuttal is the computational complexity ($2^n$) of the proposal, and the adhoc choice made by the authors to restrict to $n=12 = O(1)$ inputs.

Then there is the issue huge unjustified claims, elusive definitions, etc.

**Reviewer Scores:**

All reviewers would have kept their original scores at best. Looking at the discussion, I don't think any of the reviewers would have said, "cool, that's awesome and answers my concerns". The main issues are too deep.

---

### Decision · Program_Chairs · 2026-01-26

Reject